# Bioorthogonal in situ assembly of nanomedicines as drug depots for extracellular drug delivery

Ziyang Cao [1,2], Dongdong Li[2], Liang Zhao[2,3], Mengting Liu[2], Pengyue Ma[1], Yingli Luo[3,4] & Xianzhu Yang [1,2,3,4 ✉]

Developing precise nanomedicines to improve the transport of anticancer drugs into tumor tissue and to the final action site remains a critical challenge. Here, we present a bioorthogonal in situ assembly strategy for prolonged retention of nanomedicines within tumor areas to act as drug depots. After extravasating into the tumor site, the slightly acidic microenvironment induces the exposure of cysteine on the nanoparticle surface, which subsequently undergoes a bioorthogonal reaction with the 2-cyanobenzothiazole group of another neighboring nanoparticle, enabling the formation of micro-sized drug depots to enhance drug retention and enrichment. This in situ nanoparticle assembly strategy remarkably improves the antimetastatic efficacy of extracellular-targeted drug batimastat, and also leads to the simultaneous enhanced retention and sustained release of multiple agents for combined cocktail chemoimmunotherapy to finally elicit a potent antitumor immune response. Such in situ assembly of nanomedicines represents a generalizable strategy towards extracellular drug delivery and cocktail chemoimmunotherapy.

[1] Department of General Surgery, Guangzhou First People's Hospital, South China University of Technology, 510006 Guangzhou, P. R. China. [2] School of Biomedical Sciences and Engineering, South China University of Technology, Guangzhou International Campus, 511442 Guangzhou, P. R. China. [3] National Engineering Research Center for Tissue Restoration and Reconstruction, and Key Laboratory of Biomedical Engineering of Guangdong Province, South China University of Technology, 510006 Guangzhou, P. R. China. [4] Key Laboratory of Biomedical Materials and Engineering of the Ministry of Education, and Innovation Center for Tissue Restoration and Reconstruction, South China University of Technology, 510006 Guangzhou, P. R. China. ✉email: yangxz@scut.edu.cn

The successful transport of anticancer agents to the tumor area and final therapeutic action site is critical for successful cancer treatment[1,2]. Nanoparticle (NP)-based drug delivery systems have exhibited promising potency in the delivery of these agents into tumor tissue[3–5]. To date, several anticancer NP-based drugs have been clinically approved, and more are undergoing clinical trials[6,7]. Despite great progress, the disappointing status quo is that numerous NP-based drugs fail to improve the survival of patients in clinical trials[8,9].

Recently, Wilhelm et al. analyzed 117 published preclinical studies and reported that only 0.7% of the injected dose of NP-based drugs was delivered to the solid tumor, which is somewhat unexpected and disappointing[10]. In contrast, meta-analyses performed by Lauren et al. demonstrated that the overall exposure of nano-based drugs to a tumor was 76.12% that of the overall plasma exposure, which is a promising result[11]. These inconsistent results suggest that even though NPs allow the delivery of a greater amount of total drug to the tumor, only a small portion of drugs can ultimately be retained at the tumor site[12,13]. Hence, different approaches that both enable the enhanced retention of nanomedicines in the tumor area[14–16] and, more importantly, can further transport the cargos to their action sites[17–20] are highly desirable, and further extensive efforts should be devoted to this topic.

Here, we report a bioorthogonal in situ assembly strategy for the prolonged retention of nanomedicines within the tumor site. Through rational design, neighboring NPs with bioorthogonal groups[21–24] on their surface undergo a tumor acidity-triggered cycloaddition crosslinking reaction (Fig. 1a), which causes a sudden size enlargement to form microscale particles. As a result, this in situ assembly strategy resulted in the formation of drug depots in the tumor area, significantly prolonging the retention of anticancer agents in tumor sites (Fig. 1b). Owing to these features, this strategy can effectively improve the antimetastatic efficacy of batimastat (BB94; an inhibitor targeting extracellular matrix metalloproteinases (MMPs)[25–27]) via a reduction in the number of pulmonary metastatic nodules in an orthotopic 4T1 tumor model. In addition, such an in situ NP assembly strategy also leads to the enhanced retention and sustained release of multiple types of drugs, including chemotherapeutics and immunotherapy agents, maximizing the performance of cocktail chemoimmunotherapy and generating remarkably improved cancer treatment.

## Results

### Design and characterization of the bioorthogonal in situ-assembled NPs in the acidic tumor microenvironment.
Figure 1 illustrates the rational design of the bioorthogonal cycloaddition reaction-mediated in situ crosslinking and assembly of the NPs in the acidic tumor microenvironment for enhanced tumor retention. Generally, the bioorthogonal reactive group cysteine (Cys) or 2-cyanobenzothiazole (CBT)[21,22] functioned as the terminal groups of the diblock polylactic acid (PLA) and polyethylene glycol (PEG) polymer (Supplementary Figs. 1–6), and the obtained polymers were noted as Cys-PEG-b-PLA and CBT-PEG-b-PLA, respectively. The second-order rate constant for the reaction of Cys and CBT was $9.19\,M^{-1}\,s^{-1}$, which is significantly larger than the value of classic biocompatible click reaction between azides and alkynyl ($7.6 \times 10^{-2}\,M^{-1}\,s^{-1}$)[22,28]. To achieve a specific bioorthogonal reaction in the tumor site, the cysteine residues of Cys-PEG-b-PLA were modified with 2,3-dimethylmaleic anhydride (DA) to obtain DA-Cys-PEG-b-PLA (Supplementary Figs. 7 and 8). The resultant DA-Cys-PEG-b-PLA and CBT-PEG-b-PLA were employed to prepare the NPs, which were denoted as D-NP and C-NP, respectively. Dynamic light scattering (DLS) and transmission electron microscopy (TEM) analyses indicated that both D-NP and C-NP had a sphere-like morphology with a similar size of ~65 nm (Fig. 2a, b). Additionally, D-NP, C-NP, and their mixture of D-NP and C-NP exhibited colloidal stability in serum (Supplementary Fig. 9).

According to our design, the amide bond between the cysteine residues and DA can be cleaved in the acidic environment[29,30] of the tumor to expose a free 1,2-aminothiol group on the cysteine residues, which can trigger the bioorthogonal reaction with the cyano moiety of the CBT structure (Fig. 2c). This leads to the in situ assembly of the extracellular drug depots for the long retention and sustained release of anticancer agents in the tumor site. To verify the bioorthogonal reaction-mediated crosslinking between D-NP and C-NP under acidic conditions, we examined particle morphology changes. After treatment at pH 6.5 for 24 h, TEM and DLS analyses (Fig. 2d, e) demonstrated gradual in situ particle assembly in the mixture solution of D-NP and C-NP (denoted as D-NP/C-NP), and the size of the particles increased to $2611 \pm 115$ nm after 24 h. In sharp contrast, the size of the particles in the D-NP/C-NP solution at pH 7.4 remained unchanged even after 24 h. Furthermore, we evaluated the influence of NP concentration on the in situ assembly effect (Fig. 2f). The images clearly demonstrate that the solution of D-NP/C-NP became turbid after incubation at pH 6.5 at all NP concentrations tested, while this phenomenon was not found at pH 7.4, thus confirming that the in situ assembly of D-NP/C-NP at pH 6.5 occurred regardless of NP concentration. In addition, a new peak at 410 nm in the fluorescence spectrum emerged after D-NP/C-NP coincubation at pH 6.5, indicating the formation of a firefly luciferin structure[31] (Fig. 2g). Moreover, the bioorthogonal reaction was accompanied by the chemical coupling between Cys-PEG-b-PLA and CBT-PEG-b-PLA, which was then evaluated by gel permeation chromatography (GPC) and [1]H NMR analysis. As shown in Fig. 2h, a new peak with an elution time of approximately 31.8 min was observed after coincubation at pH 6.5. In addition, a new peak (chemical shift 2.74 ppm) appeared in the [1]H NMR spectrum, which corresponds to the methylene hydrogen atom of the cysteine residue after the covalent click cycloaddition reaction (Supplementary Fig. 10).

Moreover, to directly monitor particle aggregation, DiI was used to label D-NP and C-NP, and then both NPs were mixed and incubated at pH 6.5 for 3D confocal laser scanning microscopy (CLSM) analysis. As shown in Fig. 2i, microscale particles gradually emerged, and their number and size increased with increasing treatment time, which clearly confirmed that D-NP and C-NP could undergo in situ assembly under acidic conditions and form microscale aggregates.

### Bioorthogonal in situ assembly of NPs as drug depots within the tumor area improved the therapeutic efficacy of extracellular-targeting anticancer drugs.
To investigate whether the bioorthogonal in situ assembly of NPs would induce their long-term retention and enrichment in the tumor site, orthotopic 4T1 tumor-bearing mice were administered DiD-labeled D-NP/C-NP intravenously (i.v.) and then monitored using a fluorescence in vivo imaging system (IVIS). In addition, the acid-insensitive control polymer SA-Cys-PEG-b-PLA (S-NP) (Supplementary Fig. 11), which was comprised of the Cys-PEG-b-PLA polymer modified with succinic anhydride and also shown good colloidal stability in serum (Supplementary Fig. 12), was labeled to form DiD-labeled S-NP/C-NP for i.v. injection as the control formulation. As shown in Fig. 3a, mice treated with D-NP/C-NP or S-NP/C-NP exhibited similar and enhanced fluorescence signals at the tumor site at 4 h, 8 h, and 12 h post-injection, which could be due to the enhanced permeability and retention effect of the nanomedicine. However, as

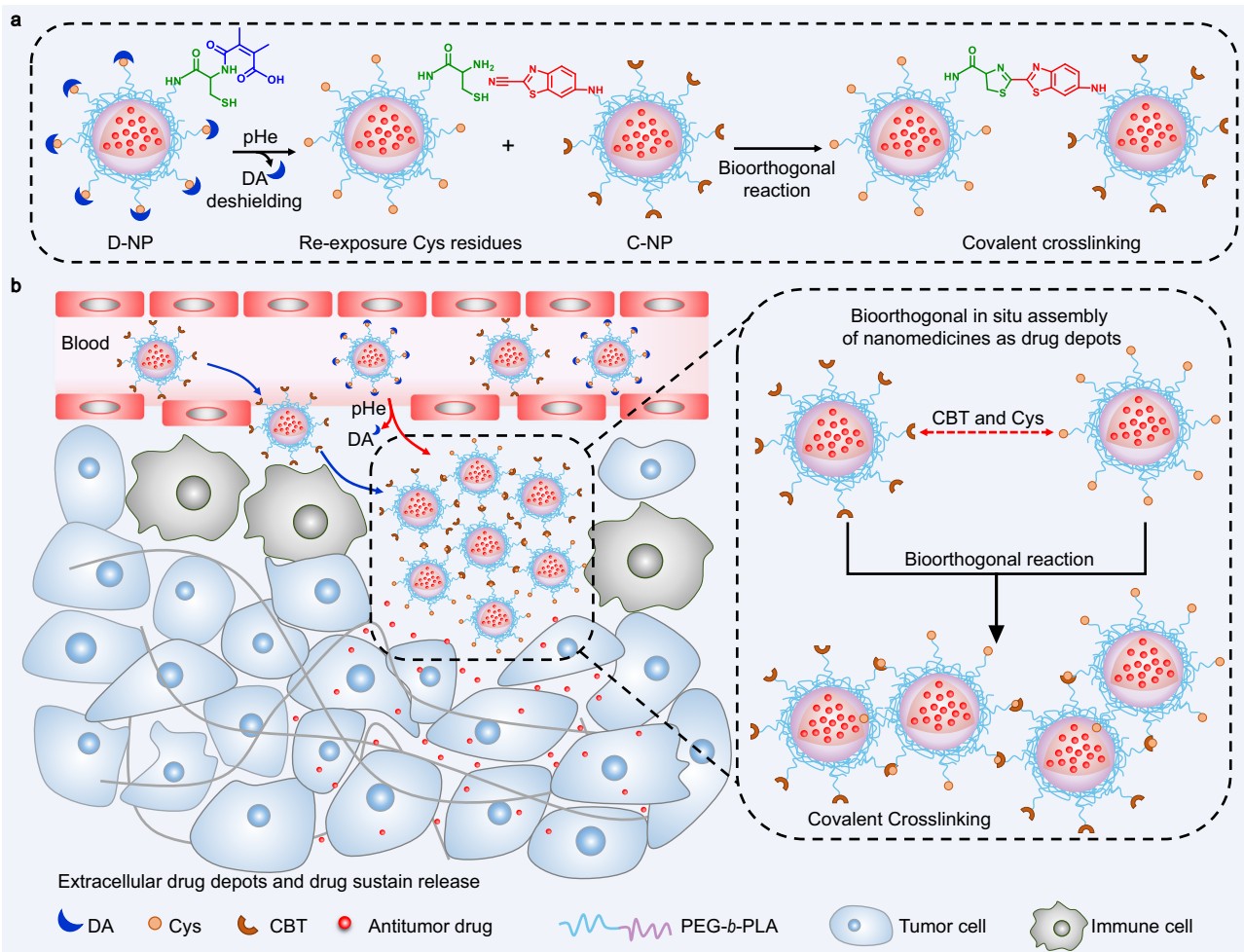

**Fig. 1 Schematic illustrating the bioorthogonal in situ assembly strategy to construct drug depots within tumor areas. a** Two types of NPs, D-NP and C-NP, undergo a bioorthogonal crosslinking reaction in the acidic tumor microenvironment. The cysteine residues of D-NP are re-exposed under the acidic conditions and can react with the CBT residues of C-NP. **b** The bioorthogonal in situ-assembled combination of D-NP and C-NP within the tumor area could serve as a drug depot to prolong the retention of the nanomedicine in the tumor site and for sustained release of the anticancer agents for cancer therapy. DA: 2, 3-dimethylmaleic anhydride, pHe: tumor acidity, Cys: cysteine, CBT: 2-cyanobenzothiazole, PEG-*b*-PLA: polyethylene glycol-polylactic acid block copolymer.

time went on, the mice in the D-NP/C-NP group exhibited remarkably enhanced fluorescence signals in the tumor site compared to that in the control S-NP/C-NP group at 24–96 h post-injection. Moreover, ex vivo imaging at 96 h post-injection clearly indicated the most intense fluorescence signals in the tumor tissues of the mice in the D-NP/C-NP group (Fig. 3b). Quantitative region-of-interest (ROI) analysis showed that the fluorescence intensity of the tumor tissue of mice treated with D-NP/C-NP was approximately 2.6-fold higher than that in the S-NP/C-NP group (Fig. 3c).

Furthermore, we used intravital CLSM to visualize the in situ assembly and long retention of the NPs in the tumor in real time. Mice bearing an orthotopic GFP-4T1 tumor with dorsal skinfold window chambers were i.v. injected with Cy5-labeled D-NP/C-NP or the control formulation S-NP/C-NP, and then the NPs retention behavior was observed by intravital CLSM (Fig. 3d). As shown in Fig. 3e and Supplementary Movies 1 and 2, the red fluorescence signals of both D-NP/C-NP and S-NP/C-NP were confined in the blood vessels at 10 min post-injection followed by gradual penetration into the green GFP-4T1 tumor tissue. Notably, in the D-NP/C-NP treatment group, many large red fluorescence aggregates were observed in the tumor site at 6 h post-injection (Fig. 3e), which was not the case in the mice treated with S-NP/C-NP. More importantly, the red aggregates in the

mice in the D-NP/C-NP group remained in the tumor site even at 48 h post-injection. To further measure the precise biodistribution, a hydrophobic platinum prodrug (Supplementary Fig. 13) was chosen as a model drug to facilitate quantitative analysis by inductively coupled plasma mass spectrometry (ICP-MS). The D-NP/C-NP and S-NP/C-NP showed comparable pharmacokinetics curve in blood and similar accumulation in the major organs liver, kidney, spleen, lung, and heart (Supplementary Fig. 14a–f). However, the platinum contents in tumor tissues in D-NP/C-NP group was obviously higher than that of S-NP/C-NP group at 48 h to 96 h post-injection (Supplementary Fig. 14g). In addition, the D-NP/C-NP and S-NP/C-NP were covalently labeled by Cy5 to determine their biodistribution (Supplementary Fig. 15), which also confirmed the enhanced tumor accumulation of D-NP/C-NP group. These results collectively confirmed the efficient in situ assembly of D-NP/C-NP in the tumor through the tumor acidity-triggered bioorthogonal reaction, resulting in the NPs showing significantly prolonged retention and enrichment in the tumor sites to serve as drug depots for the sustained release of anticancer drugs.

To demonstrate the advantage of such an intratumoral in situ assembly strategy, we selected BB94 as a model extracellular-targeted drug to evaluate the antitumor efficiency of D-NP/C-NP.

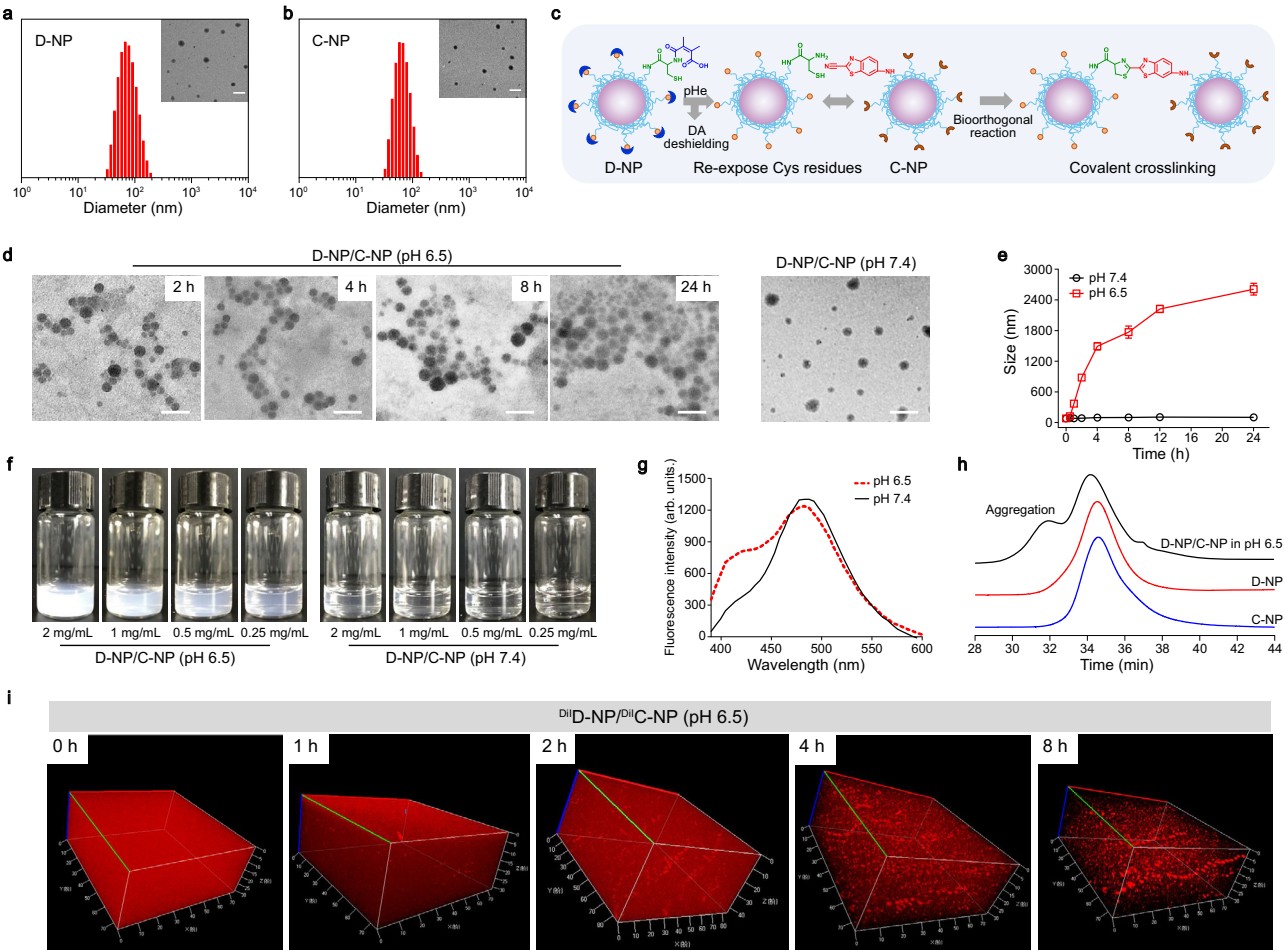

**Fig. 2 Construction and characterization of the bioorthogonal in situ-assembled nanomedicines under acidic conditions. a, b** Average hydrodynamic sizes of **a** D-NP and **b** C-NP determined by DLS, demonstrating their similar size of approximately 65 nm ($n = 3$). The inset images are representative TEM images of D-NP and C-NP, Scale bar, 200 nm. Experiments were repeated twice. **c** Schematic illustrating the bioorthogonal in situ assembly of D-NP/C-NP in the acidic tumor microenvironment. **d** TEM images of D-NP/C-NP incubated under acidic conditions for various lengths of time. D-NP/C-NP incubated at pH 7.4 for 24 h was used as a control formulation. Scale bar, 200 nm. Experiments were repeated twice. **e** DLS measurements of the hydrodynamic diameters of D-NP/C-NP after incubation under acidic conditions for various lengths of time. Data are presented as the mean ± s.d. ($n = 3$). **f** Photographs of the D-NP/C-NP aqueous solutions after incubation at pH 6.5 or 7.4 for 2 h at various NP concentrations. Experiments were repeated twice. **g** Fluorescence spectra of the D-NP/C-NP aqueous solutions after incubation at pH 6.5 or 7.4 for 2 h. Experiments were repeated twice. **h** GPC analysis of the freeze-dried D-NP/C-NP after coincubation at pH 6.5 for 2 h. Freeze-dried D-NP and C-NP were used as control formulations. Experiments were repeated twice. **i** 3D CLSM images of the D-NP/C-NP aqueous solutions at pH 6.5 after incubation for various lengths of time. D-NP and C-NP were labeled with DiI. A representative image from one of five independent fields of view in a single experiment. DA: 2, 3-dimethylmaleic anhydride, pHe: tumor acidity, Cys: cysteine, Arb. units.: arbitrary units, DiI: 1,1'-Dioctadecyl-3,3,3',3'-tetramethylindocarbocyanine perchlorate. Source data are provided as a Source Data file.

BB94 is a potent extracellular broad-spectrum MMP inhibitor that inhibits MMP-1, MMP-2, MMP-3, MMP-7, and MMP-9 and distant tumor metastases. After encapsulation, BB94 showed a gradual and sustained release behavior for both D-NP$_{BB94}$/C-NP$_{BB94}$ and S-NP$_{BB94}$/C-NP$_{BB94}$ (Supplementary Fig. 16). Notably, in situ assembly of D-NP/C-NP under acidic conditions can efficiently decrease the cellular uptake due to the inefficient internalization of the formed microscale aggregates (Supplementary Fig. 17, Supplementary Fig. 18 and Supplementary Movies 3), which could improve the delivery efficacy of BB94 into the extracellular MMP. Further, we measured extracellular and intracellular distribution of these nanoparticles and BB94 in vivo (Supplementary Fig. 19). It could be clearly observed that D-NP/C-NP treatment could significantly enhance extracellular contents of NPs and BB94 compared to the S-NP/C-NP group, especially at 24 h and 48 h post-injection.

Subsequently, the anticancer activities of BB94-loaded D-NP$_{BB94}$/C-NP$_{BB94}$ and S-NP$_{BB94}$/C-NP$_{BB94}$ were evaluated in mice bearing 4T1 breast orthotopic tumors. The 4T1 tumor-bearing mice were i.v. administered NPs on days 1, 3, and 5 at a BB94 dosage of 5.0 mg/kg body weight (Fig. 4a), and the lungs of these treated mice were collected at day 27 to evaluate therapeutic efficacy. Tumor lung metastasis was obvious in the free BB94 group, and the S-NP$_{BB94}$/C-NP$_{BB94}$ group exhibited moderate lung metastasis inhibition compared with mice treated with PBS (Fig. 4b, c). In contrast, D-NP$_{BB94}$/C-NP$_{BB94}$ treatment showed significant therapeutic efficacy, with a 93% reduction in the number of pulmonary metastatic nodules compared to the PBS group (Fig. 4c). In addition, D-NP$_{BB94}$/C-NP$_{BB94}$ treatment also efficiently suppressed the primary orthotopic 4T1 breast tumor growth (Supplementary Fig. 20). More importantly, western blot and tumor immunohistochemical staining assays demonstrated

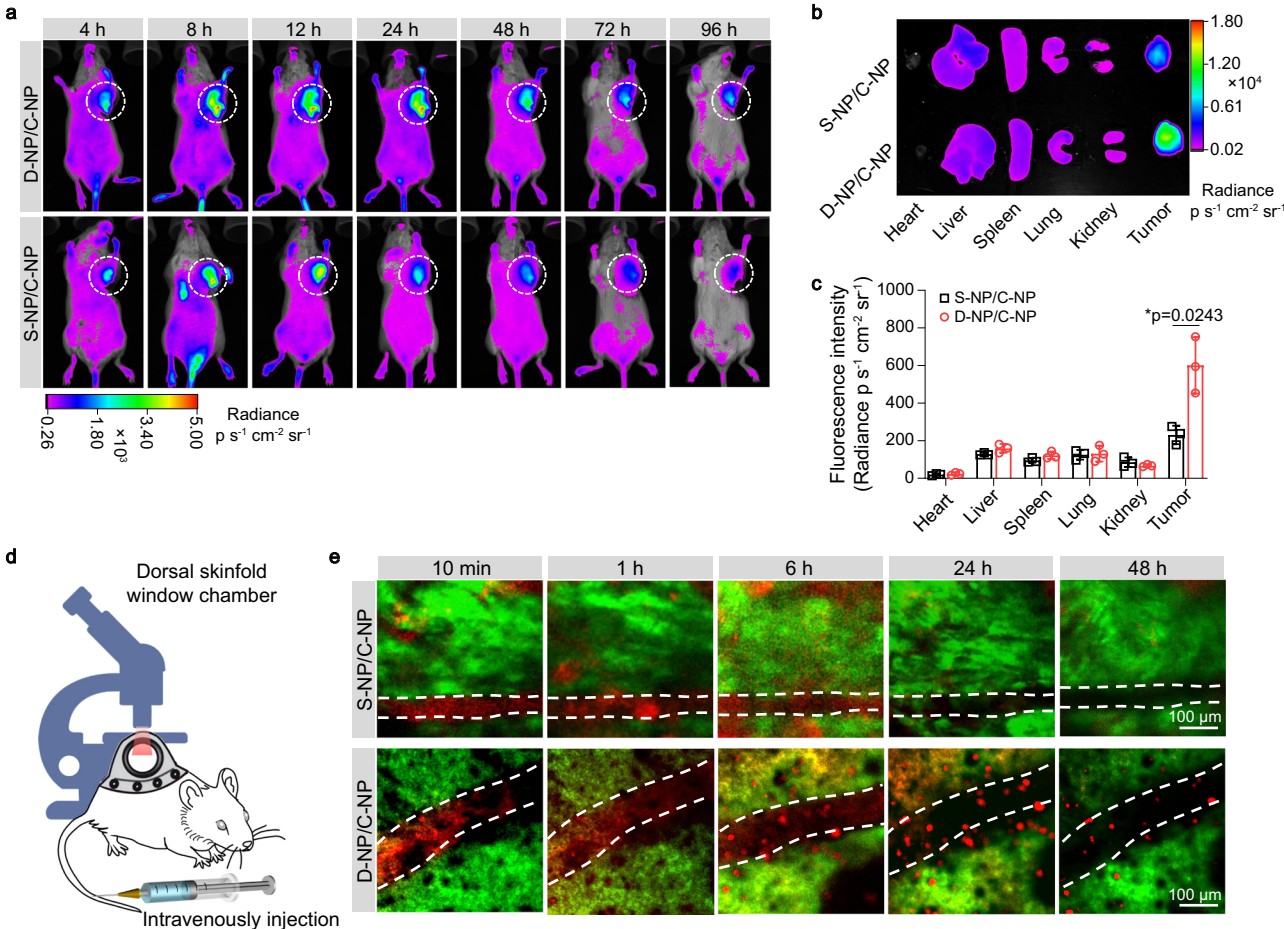

**Fig. 3 Intratumoral in situ assembly of D-NP/C-NP achieves long retention and enrichment of the NPs in the tumor site to serve as drug depots. a** IVIS imaging of 4T1 tumor-bearing mice at different times after i.v. injection of the mixture of D-NP plus C-NP (D-NP/C-NP) or S-NP plus C-NP (S-NP/C-NP) ($n = 3$). The NPs were labeled with DiD. White circles indicate the tumor sites. **b** Representative ex vivo imaging of the tumor tissues and major organs after i.v. injection of D-NP/C-NP or S-NP/C-NP at 96 h post-injection. **c** ROI analysis of the fluorescent signals of the collected tumor tissue and major organs in **b**. Data are presented as the mean ± s.d. ($n = 3$). Statistical significance was calculated via Student's *t*-test (two-tails) in **c**. *$p < 0.05$. **d** Schematic showing the use of intravital CLSM to visualize the intratumoral in situ assembly and long retention of D-NP/C-NP in real time. **e** Intravital CLSM real-time visualization of D-NP/C-NP and S-NP/C-NP tumor accumulation in GFP-4T1 tumor-bearing mice. Both D-NP/C-NP and S-NP/C-NP were labeled with Cy5. A representative image from one of five independent fields of view in a single experiment. Source data are provided as a Source Data file.

decreased expression of MMP-2, MMP-3, and MMP-9 in the primary 4T1 tumor after D-NP$_{BB94}$/C-NP$_{BB94}$ treatment (Fig. 4d–f), which further confirmed its excellent therapeutic activity. In addition, D-NP$_{BB94}$/C-NP$_{BB94}$ reduced the systemic toxicity of BB94 to 4T1 tumor-bearing mice, as evidenced by body weight measurements (Supplementary Fig. 21).

**Intratumoral bioorthogonal in situ assembly of NPs delivered multiple agents simultaneously for efficient cocktail chemoimmunotherapy.** For the above-described intratumoral in situ assembly of NPs as drug depots, its components D-NP and C-NP could transport different types of anticancer agents to simultaneously deliver multiple agents via the individual NPs (Fig. 5a). To verify this hypothesis, D-NP and C-NP were labeled with DiI and DiD, respectively, and then coincubated at pH 6.5. As expected, 3D CLSM analysis demonstrated large aggregates formed after 2 h of coincubation (Fig. 5b). More importantly, the fluorescence signals of DiI and DiD displayed good colocalization, indicating the capability of the D-NP/C-NP formulation to deliver multiple agents simultaneously.

Moreover, we examined the in vivo simultaneous delivery potency of the D-NP/C-NP formulation via i.v. injection into 4T1 tumor-bearing mice. At 96 h post-injection, frozen tumor sections were stained with DAPI and imaged by CLSM. The results showed strong colocalization of the DiI and DiD signals in the D-NP/C-NP group (Fig. 5c), but this colocalization was negligible in the S-NP/C-NP-treated group. More interestingly, both the DiI and DiD signals in the tumor tissue in the D-NP/C-NP group were more intense than those in the S-NP/C-NP group, suggesting the enhanced retention and enrichment of NPs in the tumor site via bioorthogonal in situ assembly within the tumor area. To confirm this result, the contents of both NPs in tumor tissue were quantitatively determined after i.v. injection. Within the first 12 h after treatment, both formulations exhibited comparable retention in the tumor tissues (Fig. 5d, e), which is corresponded with their particle clearing kinetics in blood circulation (Supplementary Fig. 22). Subsequently, the contents of Cy5 and FITC decreased rapidly in the S-NP/C-NP group, whereas the Cy5- and FITC-labeled NPs were retained in the tumor site of D-NP/C-NP-treated mice. The final tumor contents of Cy5- and FITC-labeled NPs in D-NP/C-NP-treated mice were 8.63 ± 3.93 µg/g and 10.16 ± 4.20 µg/g of tumor tissue at 96 h post administration, respectively, which were 1.92-fold and 1.93-fold higher than those of S-NP/C-NP-treated mice (Fig. 5d, e).

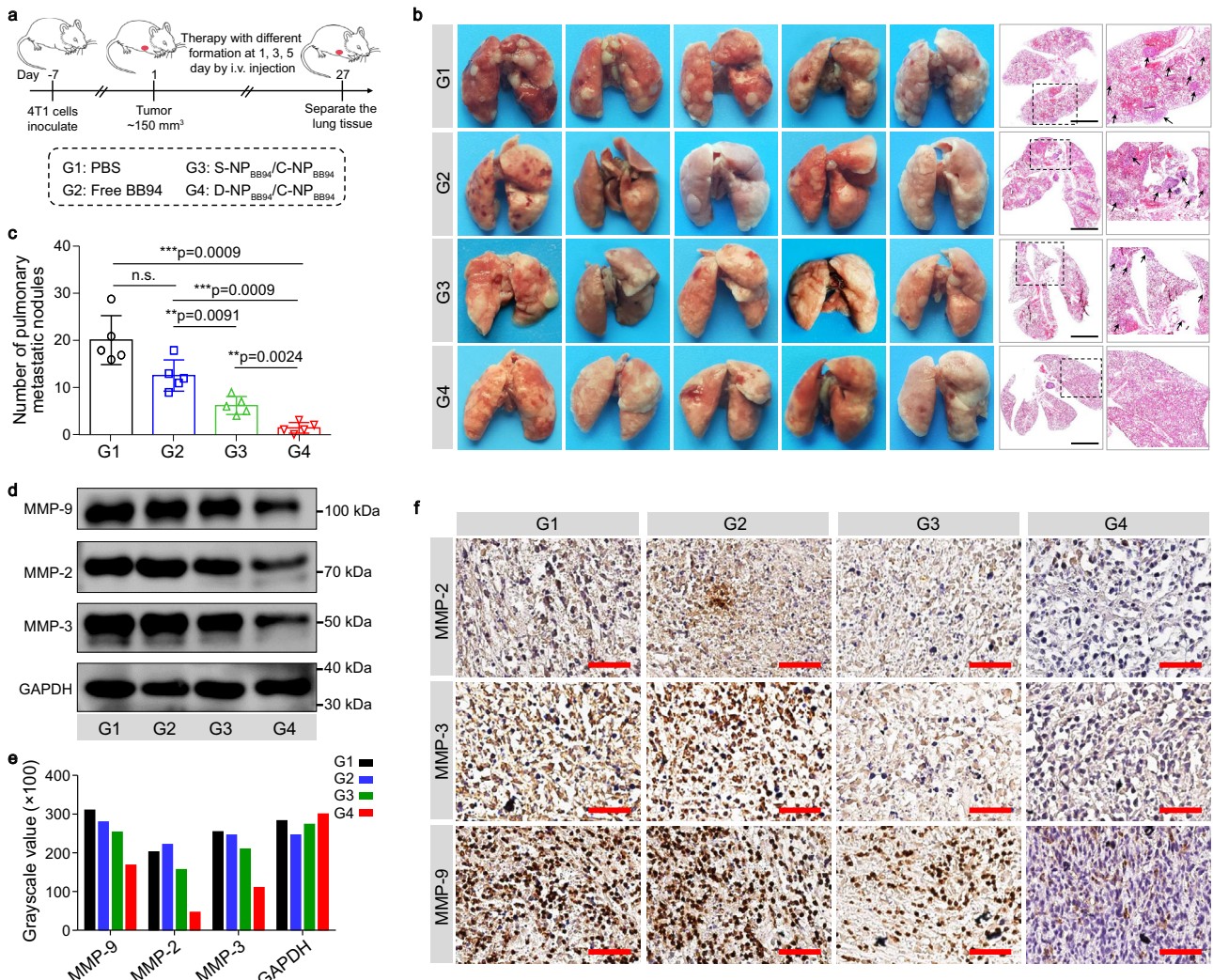

**Fig. 4 The bioorthogonal in situ nanomedicine assembly enhanced the anticancer activity of BB94. a** Schematic illustrating the experimental protocol for the orthotopic 4T1 breast tumor model. **b** Images and H&E staining of collected lung tissues at the end of treatment ($n = 5$). Scale bar, 0.5 cm. **c** The number of metastatic lung nodules from mice bearing 4T1 tumors at the end of treatment. Data are presented as the mean ± s.d. ($n = 5$). Statistical significance was calculated via Student's $t$-test (two tail). ** $p < 0.01$, *** $p < 0.001$; n.s., no significant difference. **d** Immunoblot analysis of MMP-2, MMP-3, and MMP-9 in primary tumor tissue at the end of treatment. Experiments were repeated twice. **e** Statistical analysis of the stripe grayscale values in **d**. **f** Tumor immunohistochemical staining of MMP-2, MMP-3 and MMP-9 from primary tumor of each group at the end of treatment. Scale bar, 100 μm. A representative image from one of five independent fields of view in a single experiment. i.v.: intravenously, PBS: phosphate buffered saline, MMP: matrix metalloproteinase protease. Source data are provided as a Source Data file.

The increased colocalization and more intense fluorescence signals present in the tumors confirmed efficient intratumoral in situ assembly of these NPs, which could serve as drug depots to improve the delivery of multiple agents to the tumor region and lengthen their retention.

To evaluate the potency of D-NP/C-NP as drug depots to deliver multiple agents for cocktail therapy, we chose doxorubicin (DOX), NLG919 and BLZ945 as model drugs. DOX is a chemotherapeutic[32,33], and NLG919 and BLZ945 are classical immunotherapy agents that inhibit indoleamine 2,3-dioxygenase 1 (IDO-1)[34–36] and colony-stimulating factor 1 receptor[37,38], respectively. DOX was encapsulated into D-NP (D-NP$_{DOX}$) or S-NP (S-NP$_{DOX}$), while NLG919 and BLZ945 were co-encapsulated into C-NP (C-NP$_{NLG\&BLZ}$); and the three drugs could gradually release from these NPs (Supplementary Fig. 23). Mice bearing orthotopic 4T1 tumor were randomly divided into six groups ($n = 5$): (i) PBS, (ii) DOX/BLZ945/NLG919, (iii) D-NP$_{DOX}$, (iv) D-NP$_{DOX}$/C-NP, (v) S-NP$_{DOX}$/C-NP$_{NLG\&BLZ}$, and

(vi) D-NP$_{DOX}$/C-NP$_{NLG\&BLZ}$. The mice received a total of two i.v. injections of the appropriate formulation on days 0 and 2 (Fig. 6a). As expected, the intratumoral in situ-assembled NPs acted as drug depots to improve the retention and sustained release of multiple agents after their delivery to the tumor region. Tumor growth was dramatically halted after two administrations of D-NP$_{DOX}$/C-NP$_{NLG\&BLZ}$ (group 6), with a tumor inhibition rate of 82.49% (Fig. 6b, c). In contrast, the mice treated with the NPs without crosslinking ability (S-NP$_{DOX}$/C-NP$_{NLG\&BLZ}$, group 5) displayed only slight tumor growth suppression, achieving 43.83% tumor inhibition. In addition, the anticancer activity of chemotherapy alone was negligible after two rounds of D-NP$_{DOX}$ treatment (group 3), but the effects of DOX were clearly augmented upon encapsulation into NPs that had the ability to assemble in situ (group 4). Additionally, these treated mice did not exhibit noticeable weight loss during treatment (Supplementary Fig. 24). Moreover, the best therapeutic effects from D-NP$_{DOX}$/C-NP$_{NLG\&BLZ}$-mediated cocktail chemoimmunotherapy were

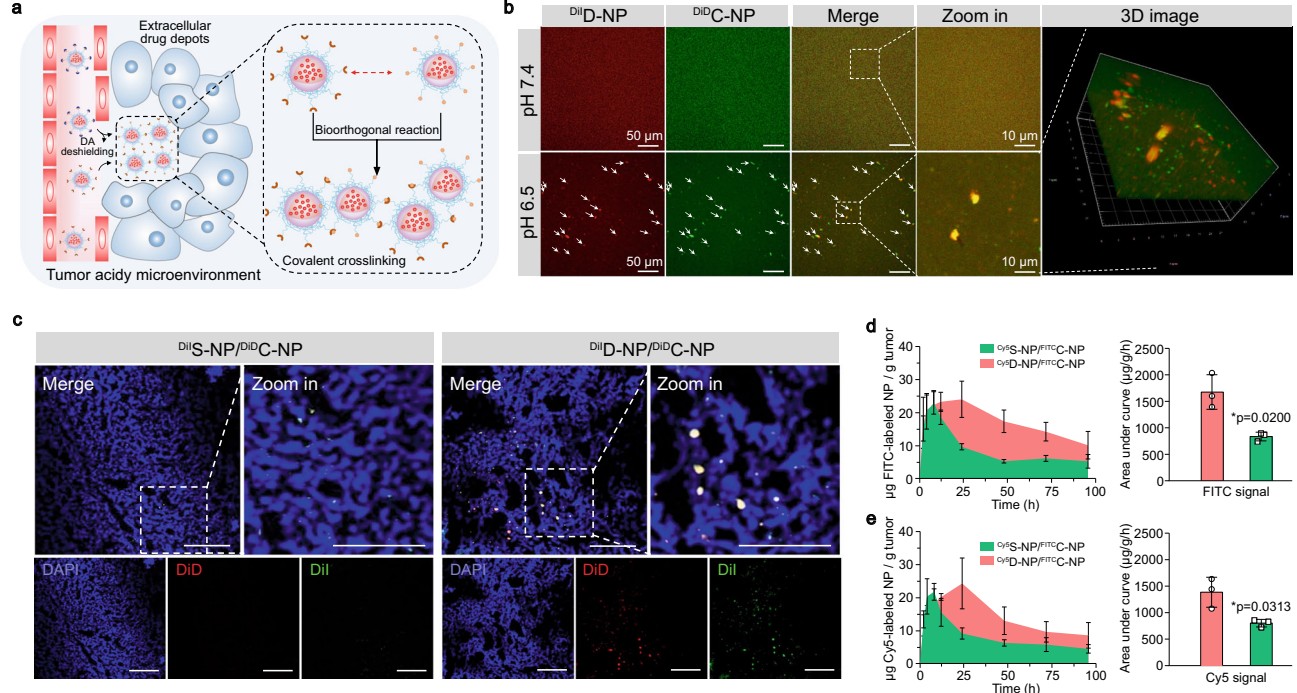

**Fig. 5 Intratumoral in situ assembly of D-NP/C-NP constructs drug depots to deliver multiple agents into the tumor site simultaneously. a** Schematic illustrating the bioorthogonal in situ assembly of D-NP/C-NP in the tumor area for the simultaneous delivery of multiple agents. **b** 3D CLSM images of the D-NP/C-NP aqueous solutions at pH 7.4 and 6.5 after incubation for 2 h. D-NP and C-NP were labeled with Dil and DiD, respectively. Experiments were repeated twice. **c** In vitro fluorescence imaging of frozen tumor sections after 96 h administration of D-NP/C-NP or S-NP/C-NP. C-NP was labeled with DiD, and D-NP and S-NP were labeled with Dil. Scale bar, 100 μm. Experiments were repeated twice. **d, e** Time-dependent tumor accumulation (left) of **d** FITC-labeled and **e** Cy5-labeled NPs after administration of D-NP/C-NP or S-NP/C-NP. C-NP was labeled with FITC, and D-NP and S-NP were labeled with Cy5. The area under the curve (AUC) (right) of the labeled NPs in tumor tissues. Data are presented as the mean ± s.d. ($n = 3$). Statistical significance was calculated via Student's $t$-test (one tail) in **d**, **e**. *$p < 0.05$. DA: 2, 3-dimethylmaleic anhydride, Dil: 1,1'-Dioctadecyl-3,3,3',3'-tetramethylindocarbocyanine perchlorate, DiD: 1,1-Dioctadecyl-3,3,3,3-tetramethylindodicarbocyanine. Source data are provided as a Source Data file.

confirmed by Ki-67 and TUNEL staining of tumor tissues (Supplementary Fig. 25). We also monitored 4T1 metastases to demonstrate the therapeutic effects of these in situ crosslinking NPs. The number of metastatic nodules in the lung was the lowest in the D-NP$_{DOX}$/C-NP$_{NLG\&BLZ}$ treatment group (Fig. 6d, e). Additionally, survival analysis also verified the best therapeutic activity of D-NP$_{DOX}$/C-NP$_{NLG\&BLZ}$ with 50% of animal survival in 48 days, while all of the mice in other groups (G1 to G5) died within 40 days (Fig. 6f).

To elucidate the mechanism of activity induced by D-NP$_{DOX}$/C-NP$_{NLG\&BLZ}$-mediated cocktail chemoimmunotherapy, we first examined the frequency of the T cell subpopulation in tumor tissues, which were collected at the end of treatment in Fig. 6c to measure tumor-infiltrating CD8$^+$ T cells (CD3$^+$CD4$^-$CD8$^+$) and immunosuppressive regulatory T cells ($T_{regs}$, CD3$^+$CD4$^+$Foxp3$^+$) by flow cytometry (Fig. 7a, b). The detailed gating strategy and the cells number were showed in Supplementary Fig. 26. The percentages of CD8$^+$ T cells in D-NP$_{DOX}$/C-NP$_{NLG\&BLZ}$-treated tumors were 2.34- and 1.74-fold higher than those in the free DOX/BLZ945/NLG919 and S-NP$_{DOX}$/C-NP$_{NLG\&BLZ}$ groups, respectively. Moreover, D-NP$_{DOX}$/C-NP$_{NLG\&BLZ}$ treatment effectively decreased the frequency of Tregs. Notably, the ratio of CD8$^+$ T cells to Tregs was the highest in the D-NP$_{DOX}$/C-NP$_{NLG\&BLZ}$ treatment group (Supplementary Fig. 27), which is consistent with this group showing the best antitumor effects among the treatments.

Furthermore, considering the altered macrophage polarization in BLZ945 treatment, the proportions of M1- and M2-type macrophages in tumor tissues were analyzed using flow cytometry. As shown in Fig. 7c and Supplementary Fig. 28, the proportion of M1-type macrophages (CD11b$^+$CD86$^+$MHC II$^+$)

was remarkably increased in the tumors of mice treated with D-NP$_{DOX}$/C-NP$_{NLG\&BLZ}$ and the proportion of M2-type macrophages (CD11b$^+$CD206$^+$) decreased after this treatment (Fig. 7d and Supplementary Fig. 29). As a result, the M1/M2 macrophage ratio (Supplementary Fig. 30) was the highest after D-NP$_{DOX}$/C-NP$_{NLG\&BLZ}$ treatment (group 6), and this result was further confirmed by immunofluorescence staining of iNOS (an M1 macrophage marker) and arginase-1 (Arg-1) (an M2 macrophage marker) in tumor sections (Fig. 7e). Besides, the immunofluorescence staining intuitively demonstrate the enhanced antitumor immune cells infiltration by D-NP$_{DOX}$/C-NP$_{NLG\&BLZ}$ treatment (Supplementary Fig. 31). These results indicated that the intratumoral in situ crosslinking strategy improved the therapeutic effects of the delivered BLZ945, and a similar effect was also observed after NLG919 delivery[33], which decreased the ratio of kynurenine/tryptophan (Kyn/Trp) following D-NP$_{DOX}$/C-NP$_{NLG\&BLZ}$ treatment (Supplementary Fig. 32). In addition, the increased secretion of cytokines, including TNF-α, IFN-γ, and IL-12p70, and the reduced concentration of IL-10 further confirmed that an effective anticancer immune response was elicited by D-NP$_{DOX}$/C-NP$_{NLG\&BLZ}$ treatment (Fig. 7f–i).

## Discussion

Currently, nanomedicines have attracted increasing attention, and a dozen NP-based anticancer agents have been approved for the treatment of solid tumors[39]. However, the unsatisfactory therapeutic efficacy of these approved NP-based anticancer agents has caused many researchers to question their utility[40,41]. Wilhelm et al. reported that only 0.7% of NPs were delivered to solid tumor tissue according to 117 published preclinical studies[10].

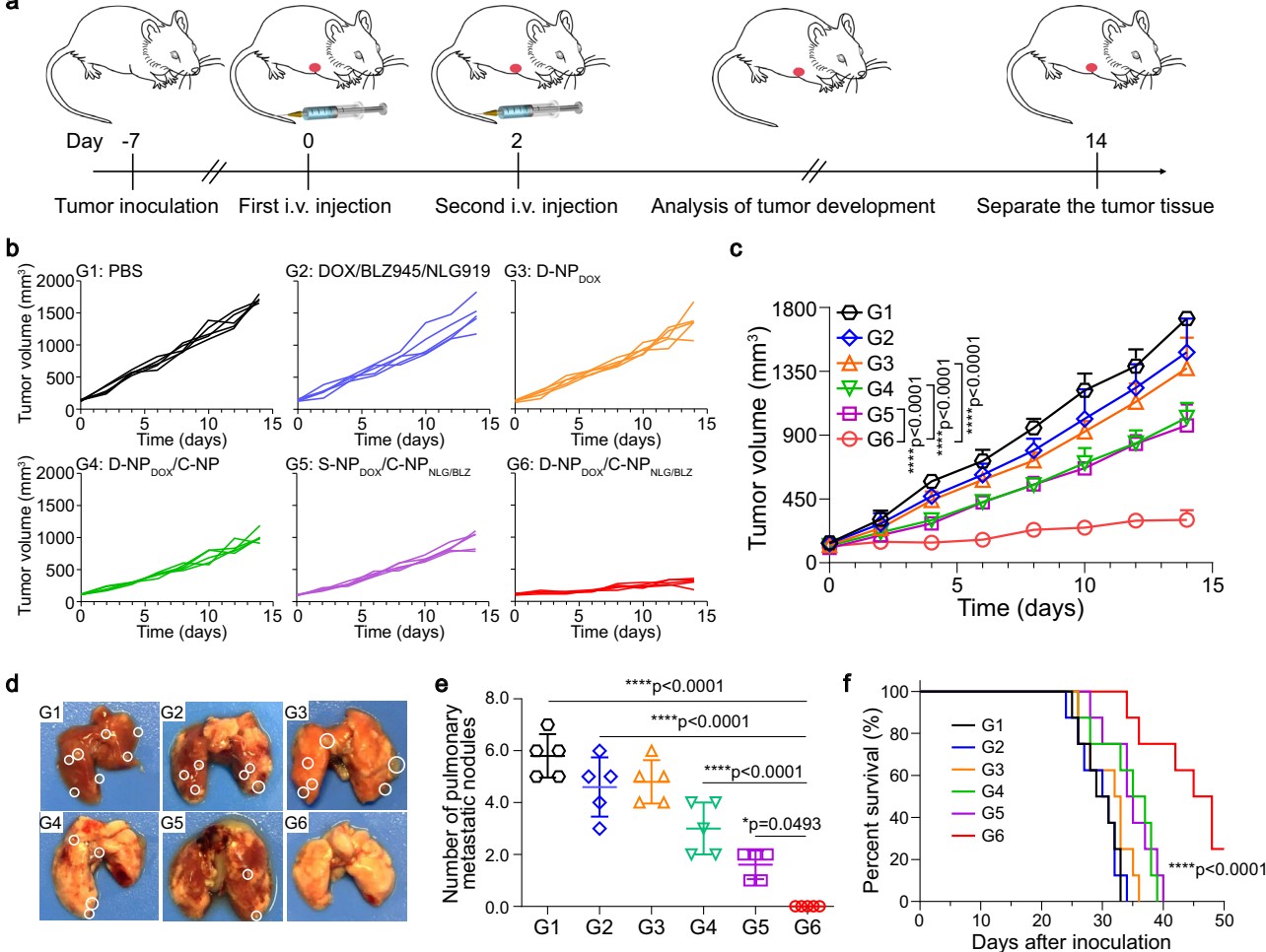

**Fig. 6 Intratumoral in situ assembly of D-NP/C-NP to deliver multiple agents for efficient cocktail chemoimmunotherapy. a** Schematic illustrating the experimental protocol for cocktail chemoimmunotherapy in the 4T1 breast tumor model. **b** Individual and **c** average growth curves of 4T1 tumors ($n = 5$). **d** Images of lung tissues collected from mice after various treatments. **e** The number of metastatic lung nodules of mice bearing 4T1 tumors at the end of treatment. Data are presented as the mean ± s.d. ($n = 5$). Statistical significance was calculated via one-way ANOVA with a Tukey post hoc test in (**c**, **e**). *$p < 0.05$, ****$p < 0.0001$. **f** Survival rates of mice treated in different groups, $n = 8$ biologically independent animals. The statistical significance of the survival was calculated via log-rank (Mantel–Cox) test. ****$p < 0.0001$. i.v.: intravenously, PBS: phosphate-buffered saline. Source data are provided as a Source Data file.

Recently, Lauren et al. queried this conclusion by analyzing the same data and found that the overall exposure of NP-based drugs in the tumor was 76.12% that of the overall exposure in plasma[11]. These inconsistent conclusions demonstrate that only a small portion of NP-based drugs can ultimately be retained in the tumor site after circulating into tumor tissue. An ongoing theme in this field of drug delivery is how to improve therapeutic efficacy, which ultimately relies on the ability of anticancer agents to reach their final site of therapeutic action.

To address the aforementioned issue, we developed D-NP and C-NP to bioorthogonally assemble in situ under acidic conditions, which is a common feature of the solid tumor microenvironment. Under acidic conditions, the cysteine residues of D-NP were re-exposed and underwent a bioorthogonal cycloaddition reaction with the CBT residues of the neighboring C-NP in the tumor area. As a result, microscale aggregates were formed via the bioorthogonal in situ assembly of the NPs, which could serve as drug depots to prolong the retention of the anticancer agent nanomedicines and sustain their release for cancer therapy. We have demonstrated the success of the bioorthogonal reaction under acidic conditions for the D-NP/C-NP formulations (Fig. 2d–h). More importantly, our extensive in vitro and in vivo

studies validated the formation of microscale drug depots via this bioorthogonal in situ assembly (Fig. 2i and Fig. 3e), which significantly enhanced the retention of the nanomedicine in tumor tissue (Fig. 3a–c).

Furthermore, we selected BB94 as the model drug to evaluate the enhancement in therapeutic efficacy due to the bioorthogonal in situ assembly strategy. Unlike common anticancer agents, BB94 is a potent extracellular inhibitor of MMP-1, MMP-2, MMP-3, MMP-7, and MMP-9, and thus, it should be released in the extracellular space rather than in tumor cells. Therefore, the bioorthogonal in situ assembly of this nanomedicine not only enhanced the retention of BB94 in the tumor tissue but also sustained drug release from the formed drug depots to ultimately improve delivery efficacy to their extracellular action sites. Thus, D-NP$_{BB94}$/C-NP$_{BB94}$ treatment exhibited remarkably improved antimetastatic efficacy (Fig. 4), with a 93% reduction in the number of pulmonary metastatic nodules compared to the PBS group. Such a bioorthogonal in situ assembly strategy can also be extended to the delivery of extracellular/membrane-targeted drugs, such as monoclonal antibodies[42–44], extracellular MMP inhibitors[45,46], tumor necrosis factor-related apoptosis-inducing ligand (TRAIL)[47,48], and melittin[49].

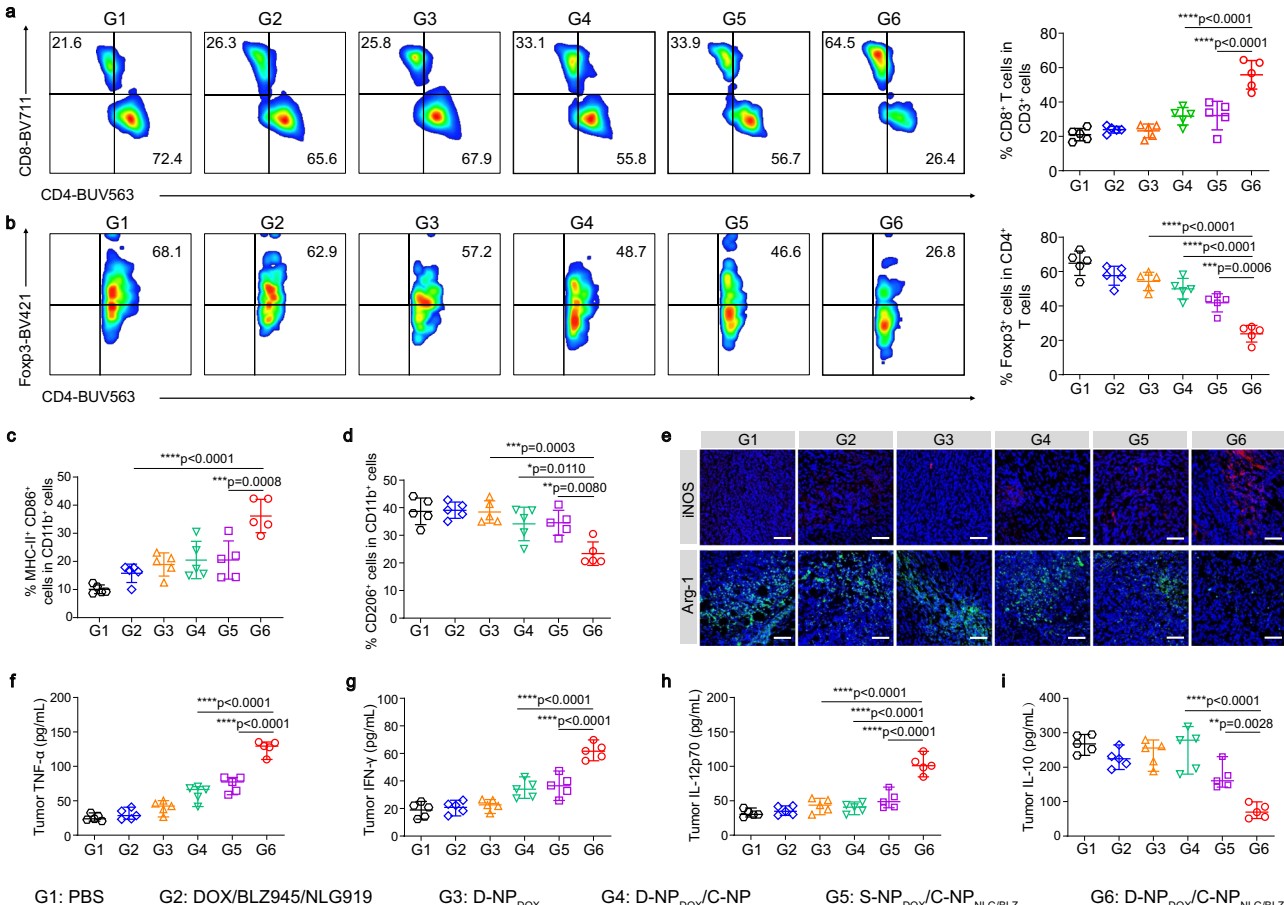

**Fig. 7 D-NP/C-NP-mediated cocktail chemoimmunotherapy relieves the immunosuppressive tumor microenvironment. a** Representative flow cytometry images (left) and relative quantification (right) of CD8+ cells. Data are presented as the mean ± s.d. (*n* = 5). **b** Representative flow cytometric analysis (left) and relative quantification (right) of Foxp3+ $T_{reg}$ cells. Data are presented as the mean ± s.d. (*n* = 5). **c, d** Relative quantification of **c** MHC II +CD86+ M1-like macrophages and **d** CD206+ M2-like macrophages in F4/80+CD11b+ macrophages. Data are presented as the mean ± s.d. (*n* = 5). **e** Representative immunofluorescence staining images for iNOS and Arg-1 in tumor sections. Scale bar, 100 μm. Experiments were repeated twice. **f–i** Levels of secreted **f** TNF-α, **g** IFN-γ, **h** IL-12p70 and **i** IL-10 in tumor tissue. Data are presented as the mean ± s.d. (*n* = 5). Statistical significance was calculated via one-way ANOVA with a Tukey post hoc test in (**a–d**, **f–i**). *$p < 0.05$, **$p < 0.01$, ***$p < 0.001$, ****$p < 0.0001$. iNOS: inducible nitric oxide synthase, Arg-1: Arginine-1, PBS: phosphate-buffered saline. Source data are provided as a Source Data file.

Moreover, in this bioorthogonal in situ assembly system, D-NP and C-NP possessed the potential to deliver different types of anticancer agents (Fig. 5) for combined therapy. To confirm this result, D-NP and C-NP were used to deliver DOX and NLG919/ BLZ945, respectively. The prepared D-NP_DOX/C-NP_NLG&BLZ dramatically inhibited tumor growth after only two administrations, reaching a tumor inhibition rate of 82.49% (Fig. 6b, c), which was clearly better than that of the control nanomedicine S-NP_DOX/C-NP_NLG&BLZ without bioorthogonal assembly capabilities. D-NP_DOX/C-NP_NLG&BLZ treatment efficiently elicited a potent antitumor immune response (Fig. 7) and ultimately achieved efficient cocktail chemoimmunotherapy for cancer therapy.

In summary, we developed a bioorthogonal in situ assembly system to enhance the therapeutic efficacies of various anticancer agents. Delivery of the extracellular-targeting MMP inhibitor BB94 with the D-NP/C-NP formulation remarkably improved the antimetastatic efficacy of BB94. In addition, considering that multiple agents could be delivered by D-NP and C-NP individually, we have proven that this bioorthogonal in situ assembly approach is a highly promising cocktail chemoimmunotherapy for cancer treatment. This bioorthogonal in situ assembly strategy provides an efficient and generalizable avenue to enhance drug delivery efficacy and opens new doors for the future successful clinical applications of nanomedicines. Notably, such a bioorthogonal in situ assembly system is based on clinical approved PEG-*b*-PLA polymeric nanoparticle, which could be helpful for future translation. However, it should also be noted that the PEG-*b*-PLA nanoparticle is usually suitable for hydrophobic drugs, which could be inefficient to encapsulate and deliver hydrophilic extracellular-target drugs such as immune checkpoint antibodies. To address this issue, an alginate nanogels system with bioorthogonal in situ assembly capability is being developed by our group.

## Methods

This research complies with all relevant ethical regulations approved by South China University of Technology.

**Antibodies.** The following flow cytometry antibodies were purchased from Bio-Legend company (San Diego, USA): CD16/32 (clone: 93), BV510 anti-mouse CD45 antibody (clone: 30-F11), BV711 anti-mouse CD8a antibody (clone: 53-6.7), APC anti-mouse CD3 antibody (clone: 17A2), BV421 anti-mouse Foxp3 antibody (clone: MF-14), PE anti-mouse CD206 antibody (clone: C068C2), BV605 anti-mouse CD11b antibody (clone: M1/70), PE/Cy7 anti-mouse F4/80 antibody (clone: QA17A29), BV650 anti-mouse CD25 antibody (clone: PC61), Percp/Cy5.5 anti-mouse Gr-1 (Ly-6G/Ly-6C) antibody (clone: RB6-8C5), APC/Cy7 anti-mouse CD86 antibody (clone: GL-1), FITC anti-mouse CD80 antibody (clone: 16-10A1),

PE/Cy5 anti-mouse CD19 (clone: 6D5), and Alexa fluor 700 anti-mouse MHC class II antibody (clone: M5/114.15.2). Flow cytometry antibodies for BUV563 anti-mouse CD4 (clone: RM4-5) was purchased from BD Company (New York, USA).

**Cell lines and animals.** The 4T1 mouse breast cancer cell line was obtained from the American Type Culture Collection (ATCC) and cultured in RPMI 1640 medium containing 10% fetal bovine serum (FBS) at 37 °C in a 5% $CO_2$ atmosphere. BALB/c mice (female, 5 weeks old) were purchased from Vital River Laboratory Animal Technology Co., Ltd. (Beijing, China). Animals were housed at approximately $22 \pm 2$ degrees centigrade; humidity $50 \pm 10\%$ on a 12 h light/ 12 h dark cycle. We performed all mice studies in accordance with the animal protocol approved by our university laboratory animal center (Approved number: AEC2017007). Experimental group sizes were approved by the regulatory authorities for animal welfare after being defined to balance statistical power, feasibility, and ethical aspects. Maximal tumor burden permitted is 2000 mm³, and the maximal tumor size/burden was not exceeded in the experiments. All animals received care in compliance with the guidelines outlined in the Guide for the Care and Use of Laboratory Animals, and all procedures were approved by the South China University of Technology Animal Care and Use Committee. The xenograft tumor model was constructed by injection of $1 \times 10^6$ 4T1 cells (100 µL) into the mammary fat pads of female BALB/c mice.

**Covalent crosslinking of D-NP and C-NP in an acidic environment.** The bioorthogonal reaction of D-NP and C-NP at pH 6.5 was verified by GPC and ¹H NMR. First, blank NPs (D-NP and C-NP) were prepared by the nanoprecipitation method. DA-Cys-PEG-*b*-PLA (10 mg) and CBT-PEG-*b*-PLA (10 mg) were weighed and dissolved in dimethyl sulfoxide (DMSO) (1 mL) and added dropwise into 10 mL of PBS (pH 7.4, 0.01 M) under stirring to obtain D-NP and C-NP, respectively. After stirring for 2 h, D-NP or C-NP was ultrafiltered with a YM-30 ultrafiltration centrifuge tube (Millipore, MWCO 3000 Da). Then, the two blank particle solutions were mixed at final D-NP and C-NP concentrations of 5 mg/mL at pH 6.5 and stirred at room temperature (R.T.) for 2 h. After that, the mixed NP solution was freeze-dried and redissolved in deuterated DMSO-$d_6$ or anhydrous dimethylformamide (DMF). After high-speed centrifugation, the supernatant was collected for ¹H NMR and GPC analyses.

D-NP and C-NP were mixed at final concentrations of 1 mg/mL at pH 6.5 and stirred for different lengths of time (2 h, 4 h, 8 h, 24 h) at R.T. At various time points, 10 µL of the mixed solution was added dropwise to copper gauze with a carbon film (400 mesh). The morphology and aggregation of the mixed particles were observed by TEM.

In addition, different concentrations of D-NP and C-NP were mixed (4 mg/mL, 2 mg/mL, 1 mg/mL, 0.5 mg/mL, and 0.25 mg/mL), the mixture pH value was adjusted to ~6.5, and the solutions stirred for 2 h at R.T. Subsequently, the turbidity of the mixed particle solutions was observed.

Then, D-NP and C-NP were mixed at pH 7.4 or pH 6.5 at a final concentration of 1 mg/mL. The mixture solutions at different pH values were stirred at R.T. for different lengths of time (0 h, 0.5 h, 1 h, 2 h, 4 h, 8 h, 12 h, and 24 h). Then, the size distribution of NPs in the solutions was detected by DLS after different stirring times.

Furthermore, fluorescence spectra of D-NP and C-NP mixed NP solutions under different pH (7.4 or 6.5) conditions were acquired with a fluorescence spectrophotometer (Ex: 280 nm, Em: 480 nm). The test solutions (D-NP and C-NP at final concentrations of 1 mg/mL) were stirred for 2 h at pH 7.4 or 6.5 before measurement.

**3D CLSM observations of D-NP and C-NP aggregation.** DiI-labeled D-NP and C-NP were prepared by a method similar to that described above while incorporating 1% DiI. DiI-labeled D-NP/C-NP was mixed at a final concentration of 1 mg/mL, and the mixture solution pH value was adjusted to ~6.5. Then, the mixture was stirred at R.T. for different lengths of time (0 h, 1 h, 2 h, 4 h, and 8 h). At different time points, the solution of DiI-labeled D-NP/C-NP aggregates was transferred into a glass-bottom cell culture dish and observed by 3D CLSM (100× oil mirror, Zeiss 880 single-photon confocal).

Next, DiI-labeled D-NP and DiD-labeled C-NP were prepared according to a method similar to that described above. Ten milligrams of DA-Cys-PEG-*b*-PLA or 10 mg of CBT-PEG-*b*-PLA was dissolved in 1 mL of DMSO and mixed with 100 µg of DiI (2 mg/mL in DMSO, 50 µL) or 100 µg of DiD (2 mg/mL in DMSO, 50 µL). Then, the mixture was added dropwise into the aqueous phase (PBS, pH 7.4, 10 mL) with stirring. The obtained ᴰⁱᴵD-NP and ᴰⁱᴰC-NP were concentrated in YM-30 ultrafiltration centrifuge tubes (Millipore, MWCO 5000 Da). Then, ᴰⁱᴵD-NP- and ᴰⁱᴰC-NP were mixed at a final concentration of 1 mg/mL. The mixed particle solution was stirred at R.T. for 2 h at pH 6.5 or 7.4. Next, the mixed particle solution was transferred to a glass-bottom cell culture dish. The aggregation between the two particles was observed by 3D CLSM (100× oil mirror, Zeiss 880 single-photon confocal).

**NP biodistribution and retention in vivo.** BALB/c female mice (7 weeks, three mice per group) bearing 4T1 xenograft tumors were i.v. injected with ᴰⁱᴵD-NP plus ᴰⁱᴰC-NP or ᴰⁱᴵS-NP plus ᴰⁱᴰC-NP (NP injection dose = 2.0 mg/mouse) and then

observed with a Xenogen IVIS® Lumina system (Caliper Life Sciences, USA) at predetermined time points. The mice were sacrificed at 96 h post-injection, and the major organs and tumor tissues were collected and imaged.

Subsequently, the tumor tissues were immersed overnight in 30% sucrose solution for immunofluorescence staining. The samples were sectioned into 8-µm-thick slides in a cryostat and briefly fixed with cold acetone. The sections were washed twice with PBS (0.01 M, pH 7.4) for 5 min at R.T. and then stained with DAPI (1 mg/mL) for 5 min. After washing with PBS, the slides were mounted with an anti-fluorescence quencher and dried overnight in the dark. Finally, the distribution of DiI- or DiD-labeled NPs in the tumor tissues was observed by CLSM.

**In vivo tumor quantification of the accumulation of D-NP plus C-NP.** 4T1 tumor-bearing female mice were randomly divided into two groups (7 weeks, 24 mice per group), the ᶜʸ⁵D-NP plus ᶠᴵᵀᶜC-NP and ᶜʸ⁵S-NP plus ᶠᴵᵀᶜC-NP groups. The mice were simultaneously i.v. administered the appropriate compounds at a dose of 2 mg of NPs per mouse. At different time points (2 h, 4 h, 8 h, 12 h, 24 h, 48 h, 72 h, and 96 h), three mice were randomly selected from each group and sacrificed. The tumor tissues were excised, washed with cold saline, dried with filter paper, weighed, and put into 2 mL FastPrep® homogenization tubes with three stainless steel beads (2 mm), and 500 µL of methanol was added. Subsequently, the tumor tissue samples were homogenized at 9391 g for 2 min, performed with 10 s of homogenization at 10 s intervals. After centrifugation (10,000 g, 10 min), the NPs in the supernatant were examined with a fluorescence spectrophotometer (Cytation™ 5, BioTek, Winooski, VT, USA). Standard curves of ᶜʸ⁵D-NP, ᶜʸ⁵S-NP and ᶠᴵᵀᶜC-NP solutions (0.8, 1.6, 3.125, 6.25, 12.5, 25, 50 and 100 µg/mL) were constructed.

**Real-time visualization of the intratumoral in situ assembly and long retention of D-NP/C-NP.** BALB/c female mice bearing GFP-4T1 tumors with dorsal skinfold window chambers were constructed according to a standard operating procedure as follows. Typically, all procedures were performed under aseptic conditions. Each recipient mouse was sedated using inhaled anesthetics (i.e., isoflurane/$O_2$) in an anesthesia chamber, transferred to a heated operating table maintained at 37 °C, and the snout was placed in the anesthesia nose cone. The hair from the entire back was removed by shaving and applying hair removal gel. Then, the centerline of the mouse spine was marked with a marker and the skin along this line was removed. Subsequently, the skinfold was pulled up, and the frames were fixed against the skin by using bolts and needles through the two small suture holes at the top of the frame. Then, a thick piece of filler glass with a diameter of 12 mm was placed and fixed with a retaining ring in the back chamber area. Finally, GFP-4T1 cells ($1 \times 10^6$ per mouse) were implanted into the transplantable window view area through the back chamber area.

BALB/c female mice bearing GFP-4T1 tumors (~80 mm³, 7 weeks) with dorsal skinfold window chambers were i.v. injected with ᶜʸ⁵D-NP plus ᶜʸ⁵C-NP or ᶜʸ⁵S-NP plus ᶜʸ⁵C-NP at a dose of 2 mg per mouse. The mice were then anesthetized by intraperitoneal injection of 125 µL pentobarbital sodium (2%, w/w) and transferred to the object stage for imaging at different times post injection (10 min, 1 h, 6 h, 24 h, and 48 h). All in vivo images were acquired using a Nikon A1RMP confocal microscope with a 20× objective. The GFP and Cy5 signals were detected using 488/520 and 633/665 nm Ex/Em filters, respectively.

**Inhibition of lung metastasis in vivo.** BALB/c female mice bearing 4T1 tumors (~150 mm³) were randomly divided into four groups (five mice per group, 7 weeks): (G1) PBS, (G2) free BB94 (2% Tween 20), (G3) S-NP$_{BB94}$/C-NP$_{BB94}$, and (G4) D-NP $_{BB94}$/C-NP $_{BB94}$. On days 1, 3, and 5, mice were i.v. injected with the indicated formulation at a BB94 dosage of 5 mg/kg body weight. Mouse weights were measured every other day. On day 27, the mice were sacrificed, the lung tissues were harvested for pulmonary metastatic nodule statistical analysis, fixed in 4% formalin solution, paraffin dehydrated, sliced, and subjected to hematoxylin and eosin (H&E) and immunohistochemical staining.

Furthermore, the tumor tissues were weighed and homogenized as described above. After centrifugation (10,000 g, 10 min), the supernatant was collected, and the expression of MMP-2, MMP-3, and MMP-9 in different tissues was analyzed by western blot according to a standard operation. The uncropped versions of immunoblot images were shown in Supplementary Fig. 33.

**In vivo antitumor performance of cocktail therapy.** BALB/c female mice bearing 4T1 tumors (~150 mm³) were randomly divided into six groups (five mice per group, 7 weeks): (G1) PBS, (G2) free DOX/NLG919/BLZ945 (denoted as DOX/NLG/BLZ), (G3) D-NP$_{DOX}$, (G4) D-NP$_{DOX}$/C-NP, (G5) S-NP$_{DOX}$/C-NP$_{NLG\&BLZ}$, and (G6) D-NP$_{DOX}$/C-NP$_{NLG/BLZ}$. On days 0 and 2, mice were i.v. injected with the appropriate formulation as described above ([DOX] = 5 mg/kg, [NLG919] = 5 mg/kg, [BLZ945] = 4.64 mg/mg). The tumor sizes and mouse weights were measured every other day, and tumor volumes were calculated by using the formula $V = L \times W^2 \times 1/2$ ($L$ is the longest dimension, and $W$ is the shortest dimension). The mean tumor inhibition rate was calculated by using the formula of $(V_P - V_t)/V_P \times 100\%$, where $V_P$ is the final tumor volume in PBS control group, and $V_t$ is the

final tumor volume at the end of different treatment groups. The mice survival times with different treatment were also monitored.

On day 14, mice were sacrificed, and the lung tissues were harvested for pulmonary metastatic nodule statistical analysis. Then, the part tumor tissues were harvested and fixed in 4% formalin solution, paraffin dehydrated, sliced and subjected to Ki-67 and TUNEL staining.

**Tumor-infiltrating lymphocyte analysis**. Immune cells present in the tumors after the above immune combination therapy were analyzed. Tumor tissues were processed through mechanical disruption, digested, and filtered to obtain single-cell suspensions, which were then stained with antibodies for flow cytometry analysis according to our previous report[50]. For instance, a single cell suspension was incubated with anti-CD16/32 (1:100 dilution) and then stained with anti-CD45-BV510 (1:100 dilution), anti-CD3-APC (1:150 dilution), anti-CD8a-BV711 (1:100 dilution), anti-CD4-BUV563 (1:200 dilution), anti-F4/80-PE/Cy7 (1:100 dilution), anti-CD25-BV650 (1:100 dilution), anti-Gr-1-Percp/Cy5.5 (1:100 dilution), anti-CD11b-BV605 (1:100 dilution), anti-CD86-APC/Cy7 (1:100 dilution), anti-CD80-FITC (1:100 dilution), anti-MHC class II-Alexa fluor 700 (1:300 dilution) and anti-CD19-PE/Cy5.5 (1:150 dilution) for 20 min at 4 °C before being fixed and permeabilized with eBioscience Foxp3/Transcription Factor Staining Buffer Set (eBioscience). Cells were then stained with anti-Foxp3-BV421 (1:100 dilution) and anti-CD206-PE (1:100 dilution) for 20 min at 4 °C and analyzed using flow cytometry (BD, LSRFortessa). The antibodies were diluted with 1×PBS containing 0.1% bovine serum albumin solution.

**Cytokine detection**. Tumor tissues were harvested and homogenized in cold PBS in the presence of digestive enzymes followed by centrifugation. The intratumor levels of IL-10 (Dakewe Biotech Co., cat# EA-2513), IL-12p70 (Dakewe Biotech Co., cat# 1211202), IFN-γ (Dakewe Biotech Co., cat# CT301A) and TNF-α (Dakewe Biotech Co., cat# CT303A) in the supernatant were measured with ELISA kits according to the manufacturer's instructions.

**Kyn and Trp in tumor tissues**. To measure Kyn and Trp in tumor tissues, the tumor tissues were dried with filter paper, weighed, and put into 2 mL FastPrep® homogenization tubes with three stainless steel beads (2 mm), and 500 μL of methanol was added. Subsequently, the tumor tissue samples were homogenized at 9391 g for 2 min, performed with 10 s of homogenization at 10 s intervals. After centrifugation (10,000 g, 10 min), the contents of Kyn and Trp in the supernatant were then examined by using a highly sensitive series triple quadrupole mass spectrometer (SCIEX, QTRAP 4500).

**Statistical analysis**. All results were expressed as mean ± s.d. Unless otherwise stated, all experiments used biological replication. One-way analysis of variance (ANOVA) was used for multiple comparisons when more than two groups were compared, and Student's $t$-test was used for two-group comparisons. The survival benefit was determined using a log-rank (Mantel–Cox) test. *$p$-values < 0.05, **$p$-values < 0.01, ***$p$-values < 0.001, ****$p$-values < 0.0001 were considered to be statistically significant. All statistical analyses were carried out with the GraphPad Prism software package (PRISM 8.0, GraphPad Prism Software) and Excel 2016.

**Reporting summary**. Further information on the research design is available in the Nature Research Reporting Summary linked to this article.

## Data availability

Source data are provided with this paper. The authors declare that all data supporting the findings of this study are available within the Article, Supplementary Information, Supplementary movies, or Source Data file. Source data are provided with this paper.

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

## Acknowledgements

This work was supported by the National Key R&D Program of China (No. 2017YFA0205600 X.Y.), the National Natural Science Foundation of China (No. 52173121 X.Y., No. 51822302 X.Y.), the Program for Guangdong Introducing innovative and Entrepreneurial Teams (No. 2017ZT07S054 X.Y.), the Natural Science Foundation for Distinguished Young Scholars of Guangdong Province (No. 2017B030306002 X.Y.). This work was also supported by the High-level Hospital Construction Project DFJH201905, and the Fundamental Research Funds for the Central Universities to X.Y.

## Author contributions

X.Y. and Z.C. were responsible for the conception and experimental strategy of the study. Z.C., D.L., L.Z., M.L., P.M., and Y.L. performed the experiments and acquired the data. Z.C., D.L., L.Z., M.L. and X.Y. interpreted the data. Z.C., D.L., L.Z, and X.Y. co-wrote the manuscript.

## Competing interests

The authors declare no competing interests.
