## [Peer Review File · Nature Communications]

REVIEWER COMMENTS

Reviewer #1 (Remarks to the Author):

This study contributed by Cao et al developed a bioorthogonal in situ assembly strategy to form the micro-sized drug depots in tumor tissue, which can substantially enhance the retention and enrichment of anticancer drugs at the tumor site. This in situ nanoparticle assembly strategy efficiently improved the therapeutic efficacy of extracellular-targeting drugs batimastat (BB94), and also delivered multiple agents simultaneously for efficient cocktail chemoimmunotherapy. In general, the idea is highly innovative, and the approach taken by the investigators is logical and systematic. Therefore, I think it's an interesting work and could meet the criteria of Nature Communications. Some issues are listed below.

1. The bioorthogonal reaction between cysteine (Cys) and 2-cyanobenzothiazole (CBT) is critical for this in situ assembly strategy. What about its reaction rate? The authors should demonstrate it, and also compare with other click reactions to explain its advantages.
2. As demonstrated in Fig. 3, the D/C-NP could form NPs aggregations to prolong retention time and achieve sustain release of BB94. However, its release profile was not shown. The authors should provide release behavior of BB94 before and after formation of particle aggregations. And, its release from the control formulation should also be studied.
3. As the D/C-NP was employed to deliver extracellular-targeting drugs BB94, the authors need to quantitatively determine the ratio of D/C-NP inside and outside the tumor cells.
4. The authors have designed in vivo animal experiments to demonstrate the bioorthogonal in situ assembly to minimize the cellular uptake and thus enhance the delivery efficacy of BB94 to extracellular matrix metalloproteinases. To clearly demonstrate it, the cellular uptake of D/C-NP could be investigated in vitro to prove that the formed aggregations could stay outside of the cells.
5. Compared to flow cytometric analysis, confocal laser scanning microscopy (CLSM) can intuitively investigate the type, density, and location of immune cells within tumors, which could be provided in the revised manuscript.
6. It would be instructive if a discussion of potential limitations and challenges of this approach in terms of future translation could be supplemented in the revised manuscript.

Reviewer #2 (Remarks to the Author):

This submission reports the use of a previously reported bioorthogonal chemical reaction between 2-cyanobenzothiazole (CBT) and free cysteine to cross link nanoparticles into microparticles in situ. The formed larger particles are expected to retain longer than nanoparticles. They designed a diblock polyacetic acid (PLA) and polyethylene glycol (PEG) polymer and functionalized the terminal groups with either CBT (which afforded C-NP) or cysteine. The cysteine-terminated polymer was further modified to afford a pH sensitive nanoparticle D-NP. The authors presented in vitro characterization of the pH dependent crosslinking of C-NP and D-NP, and then demonstrated the prolonged retention after i.v. delivery by imaging. Several drug molecules have been encapsulated in the nanoparticles and enhanced anticancer efficacy was observed with the C-NP/D-NP. Finally, the authors explored the strategy to deliver both chemo drug and immune-modulators simultaneously, which gave an impressive inhibition

of tumor growth and metastasis. Overall, the work is of high interest to those in the field of nanomedicine, and would be suitable for publication in Nature Communication after a revision on the following concerns.

1. In this strategy, the authors started with nanoparticles that still rely on EPR for tumor retention, and also non-specific RES uptake. It is unclear how those issues are addressed with this strategy. The authors should provide a more thorough biodistribution study and show what is the standardized uptake in various organs, and blood circulation half-life.
2. The proposed mechanism is that the acidic tumor microenvironment activates the cross-linking reaction. How can the authors rule out that the uptake of the nanoparticles into endosomes and subsequent crosslinking?
3. What was in situ crosslinking efficiency? Did the authors see cross-link in other tissues like liver? And what is the stability of C-NP and D-NP in blood?
4. Also how did the drug get released? Extracellularly or intracellularly? Releasing kinetics vs. particle clearing kinetics? More characterization should be provided.
5. In Fig. 7 experiments, indicate when those flow cytometry data were collected. Provide experimental details on this part, like cell numbers and gating information. Line 556 does not provide citation for previous report.
6. Line 282-284: what is the definition of tumor inhibition rate? Clarify how they are calculated.

Reviewer #3 (Remarks to the Author):

In this manuscript, the author developed a tumor acid triggered bioorthogonal system by cysteine group and 2-cyanobenzothiazole group modified nanoparticles. This technology can be used as drug depots, increasing the retention and enrichment of anticancer drugs at the tumor site. The experiments and arguments in this manuscript were well organized. Thus, I enthusiastically support the acceptance of this manuscript after addressing the following issues:

1. In Line 158, the author mentioned that "Quantitative region-of-interest (ROI) analysis showed that the fluorescence intensity of the tumor tissue of mice treated with D-NP/C-NP was approximately 2.6-fold higher than that in the S-NP/C-NP group (Fig. 3c)". We wonder that whether "2.6 fold higher" was efficient enough compared with other previous reported bioorthogonal reactions that also used for increasing the retention of anticancer drugs?
2. The experimental description of Figure 4 was not clear enough. Whether BB94-loaded D-NP/C-NP was used for a subcutaneous 4T1 tumor or an orthotopic 4T1 tumor? Why there was no tumor growth curve of the primary tumor? The results in figure 4d-f were analyzed for which kind of tumor, the primary tumor or the metastasis in lungs?
3. The drug models in figure 6 are D-NP (D-NP DOX) and C-NP (C-NP NLG&BLZ). Why was DOX loaded in D-NP alone, while NLG919 and BLZ were co-delivery in C-NP? In addition, whether the use of PLGA-PEG as the backbone for a drug carrier has limitations in the type of drug loading (such as hydrophilic and

hydrophobic), and how to adopt strategies to expand the drug loading range by such bioorthogonal reactions in the future, it is recommended to be added in the Discussion.

4. In figure 7, D-NP (D-NP DOX) and C-NP (C-NP NLG&BLZ) induced an effective anticancer immune response. To further demonstrate the long-term antitumor efficiency, the mice survival experiment of these drugs is suggested to be added.

5. The author proposed the concept of using bio-orthogonal reactions to develop drug depots. We noticed that D-NP/C-NP could load different kinds of fluorescent dyes (DiI, DiD) and drugs (BB94, DOX, NLG919, BLZ). However, all data supported the argument “enhanced drug retention in tumor site” were according to the results of fluorescent dyes. We suggest the author add another kind of real drug model, which could measure its precise content in tumor by fluorescence intensity or ICP.

Reviewer #1:

This study contributed by Cao et al developed a bioorthogonal in situ assembly strategy to form the micro-sized drug depots in tumor tissue, which can substantially enhance the retention and enrichment of anticancer drugs at the tumor site. This in situ nanoparticle assembly strategy efficiently improved the therapeutic efficacy of extracellular-targeting drugs batimastat (BB94), and also delivered multiple agents simultaneously for efficient cocktail chemoimmunotherapy. In general, the idea is highly innovative, and the approach taken by the investigators is logical and systematic. Therefore, I think it's an interesting work and could meet the criteria of Nature Communications. Some issues are listed below.

Comment 1: *The bioorthogonal reaction between cysteine (Cys) and 2-cyanobenzothiazole (CBT) is critical for this in situ assembly strategy. What about its reaction rate? The authors should demonstrate it, and also compare with other click reactions to explain its advantages.*

Response: The bioorthogonal reaction of CBT and Cys is a biocompatible, highly efficient click reaction, and its second-order rate constant reached $9.19 \text{ M}^{-1} \text{ s}^{-1}$ according to previous result of Rao et al. (*Angew. Chem., Int. Ed.*, 2009, 48, 9658), which is significantly larger than the value of classic biocompatible click reaction of azide and alkyne ($7.6 \times 10^{-2} \text{ M}^{-1} \text{ s}^{-1}$) (*Proc. Natl. Acad. Sci. USA* 2007, 104, 16793). We have discussed the advantages of bioorthogonal reaction between Cys and CBT in the revised manuscript (Line 84 to line 87). Thanks for your kind reminder.

Comment 2: *As demonstrated in Fig. 3, the D/C-NP could form NPs aggregations to prolong retention time and achieve sustain release of BB94. However, its release profile was not shown. The authors should provide release behavior of BB94 before and after formation of particle aggregations. And, its release from the control formulation should also be studied.*

Response: We would like to thank the reviewer for valuable suggestion. The release profiles of BB94 were provided in the revision (Supplementary Fig. 15), and the related description have been added in the revised manuscript (Line 209 to line 211).

Supplementary Fig. 15 The release behavior of encapsulated BB94 from S-NP_{BB94}/C-NP_{BB94} and D-NP_{BB94}/C-NP_{BB94} under different pH environment. Data were presented as the mean \pm s.d. (n = 3).

Comment 3: As the D/C-NP was employed to deliver extracellular-targeting drugs BB94, the authors need to quantitatively determine the ratio of D/C-NP inside and outside the tumor cells.

Response: According to your suggestion, we have designed a quantitative experiment to determine the ratio of D/C-NP or S/C-NP and BB94 inside and outside the tumor cells *in vivo*. The results (Supplementary Fig. 18) and the related description (Line 215 to line 218) have added in the revised manuscript.

Supplementary Fig. 18 The extracellular or intracellular quantitative analysis of **a-c** nanoparticles and **d-f** BB94 in the tumor tissue after i.v. administration Cy5-labeled S-NP_{BB94}/C-NP_{BB94} or D-NP_{BB94}/C-NP_{BB94} by fluorescence spectrophotometer and HPLC. Data were presented as the mean \pm s.d. (n = 3).

Comment 4: The authors have designed *in vivo* animal experiments to demonstrate the bioorthogonal *in situ* assembly to minimize the cellular uptake and thus enhance the delivery efficacy of BB94 to extracellular matrix metalloproteinases. To clearly demonstrate it, the cellular uptake of D/C-NP could be investigated *in vitro* to prove that the formed aggregations could stay outside of the cells.

Response: The cellular uptake of D-NP/C-NP and S-NP/C-NP have been added in the revised manuscript (FACS analysis in Supplementary Fig. 16 and CLSM observation in Supplementary Fig. 17). When the D-NP/C-NP was pre-incubated at pH 6.5 to achieve extracellular assembly and form micro-scale aggregation, their cellular uptake was

significantly decreased. In contrast, the S-NP/C-NP without in situ assembly capability exhibited more readily cellular uptake. Taken together, these *in vitro* results further verified that the formed aggregations of D/C-NP could stay outside of the cells to minimize the cellular uptake. And, the related description has been added in the revised manuscript (Line 211 to line 214).

Supplementary Fig. 16 *In vitro* assembly of D-NP/C-NP decrease the cellular uptake. **a** FACS analysis and **b** mean fluorescence intensity (MFI) of 4T1 cells incubated with Cy5-labeled D-NP/C-NP after pre-incubation at pH 6.5 for different times (0.5 h, 1 h, 2 h, 4 h). Data were presented as the mean \pm s.d. (n = 3). Statistical significance was calculated via one-way ANOVA with a Tukey post-hoc test. *p<0.05, **p<0.01, ****p<0.0001.

Supplementary Fig. 17 Series CLSM images of live 4T1 cells co-incubated with Cy5-labeled D-NP/C-NP or S-NP/C-NP. The Cy5-labeled D-NP/C-NP or S-NP/C-NP was pre-incubated for 2 h under pH 6.5. The formed particles aggregations could stay outside the cells in the $Cy5D-NP/Cy5C-NP$ group (a, b in the dashed ring), resulting in lower intracellular fluorescence signals. In contrast, higher intracellular fluorescence signals were in the $Cy5S-NP/Cy5C-NP$ group, and no obvious aggregations was formed outside the cells.

Comment 5: Compared to flow cytometric analysis, confocal laser scanning microscopy (CLSM) can intuitively investigate the type, density, and location of immune cells within tumors, which could be provided in the revised manuscript.

Response: We thank this reviewer for his/her valuable suggestion. The immunofluorescence staining (Supplementary Fig. 30) have been added in the revised manuscript to intuitively demonstrate the immune cells in the tumor tissue (Line 360 to line 362).

Supplementary Fig. 30 Representative immunofluorescence images of CD8 (green), CD4 (red) and CD206 (green) and CD86 (red) in tumor sections obtained at the end of different treatment in Fig. 6. G1: PBS, G2: DOX/BLZ945/NLG919, G3: D-NP_{DOX}, G4: D-NP_{DOX}/C-NP, G5: S-NP_{DOX}/C-NP_{BLZ&NLG}, G6: D-NP_{DOX}/C-NP_{BLZ&NLG}.

Comment 6: *It would be instructive if a discussion of potential limitations and challenges of this approach in terms of future translation could be supplemented in the revised manuscript.*

Response: The discussion has been added in the revised manuscript (Line 442 to line 448).

Reviewer #2 (Remarks to the Author):

This submission reports the use of a previously reported bioorthogonal chemical reaction between 2-cyanobenzothiazole (CBT) and free cysteine to cross link nanoparticles into microparticles in situ. The formed larger particles are expected to retain longer than nanoparticles. They designed a diblock polyacetic acid (PLA) and polyethylene glycol (PEG) polymer and functionalized the terminal groups with either CBT (which afforded C-NP) or cysteine. The cysteine-terminated polymer was further modified to afford a pH sensitive nanoparticle D-NP. The authors presented in vitro characterization of the pH dependent crosslinking of C-NP and D-NP, and then demonstrated the prolonged retention after i.v. delivery by imaging. Several drug molecules have been encapsulated in the nanoparticles and enhanced anticancer efficacy was observed with the C-NP/D-NP. Finally, the authors explored the strategy to deliver both chemo drug and immune-modulators simultaneously, which gave an impressive inhibition of tumor growth and metastasis. Overall, the work is of high interest to those in the field of nanomedicine, and would be suitable for publication in Nature Communication after a revision on the following concerns.

Comment 1: *In this strategy, the authors started with nanoparticles that still rely on EPR for tumor retention, and also non-specific RES uptake. It is unclear how those issues are addressed with this strategy. The authors should provide a more thorough biodistribution study and show what is the standardized uptake in various organs, and blood circulation half-life.*

Response: We thank this reviewer for his/her valuable suggestion. To facilitate accurate determination, a hydrophobic platinum prodrug was used as a drug model to investigate the thorough biodistribution and blood circulation of D-NP/C-NP and S-NP/C-NP *in vivo* (Supplementary Fig. 14). According to the result of Supplementary Fig. 14a, the half-life ($t_{1/2z}$) of D-NP/C-NP and S-NP/C-NP were 16.8 ± 3.8 h and 16.0 ± 2.0 h, showing comparable blood circulation. In addition, except tumor tissue, both formulations shown similar accumulation in major organs including liver, kidney, spleen, lung and heart. In tumor tissue, the accumulation of D-NP/C-NP was much higher than that of S-NP/C-NP at 48 h to 96 h post injection, which is well consistent with the IVIS imaging result in Fig. 3a.

Supplementary Fig. 14 Pharmacokinetics curve in blood and biodistribution of platinum prodrug-loaded S-NP/C-NP or D-NP/C-NP after i.v. administration to BALB/c mice bearing an orthotopic 4T1 tumor. Time course of the contents of platinum in plasma (a) and the accumulation of platinum in heart (b), liver (c), spleen (d), lung (e), kidney (f) and tumor (g) after i.v. injection at a platinum dose of 40 $\mu\text{g}/\text{mouse}$ at different time points (6 h, 12 h, 24 h, 48 h and 96 h). Data were presented as the mean \pm s.d. (n = 3). Statistical significance was calculated *via* student's t-test. *p < 0.05.

Comment 2: *The proposed mechanism is that the acidic tumor microenvironment activates the cross-linking reaction. How can the authors rule out that the uptake of the nanoparticles into endosomes and subsequent crosslinking?*

Response: We agree with the reviewer's opinion that a small part of D-NP and C-NP might be internalized by tumor cells and then cross-linked in the acidic endosomes. Indeed, it is very hard to completely avoid the cellular uptake of the nanoparticles D-NP and C-NP. However, it should be noted that the D-NP/C-NP with in situ assembly capability eventually achieved ~ 3.88 -fold BB94 contents in the extracellular space than that of S-NP/C-NP group at 48 h (Supplementary Fig. 18), which efficiently confirmed that this in situ assembly strategy is efficient to improve the delivery efficacy of extracellular-targeted drugs.

Comment 3: *What was in situ crosslinking efficiency? Did the authors see cross-link in other tissues like liver? And what is the stability of C-NP and D-NP in blood?*

Response: Thanks for your kind reminder. We approximately calculated the in situ crosslinking efficiency by the following equation:

$$\text{Crosslinking efficiency (CE\%)} = \left(1 - \frac{C_{S-NP}}{C_{D-NP}} \right) * 100\%$$

where C_{S-NP} and C_{D-NP} represent the extracellular contents of S-NP/C-NP and D-NP/C-NP nanoparticles, respectively. This equation is based on the prerequisite estimation that the same amount of nanoparticles extravasate from blood vessels into tumor tissue in the S-NP/C-NP and D-NP/C-NP group, because of their similar properties and pharmacokinetics curve. According to the result in Supplementary Fig. 18a and 18b, in situ crosslinking efficiency was approximately calculated and shown in Figure R1.

Figure R1. The estimated crosslinking efficiency of D-NP/C-NP in tumor tissues.

The acidic extracellular microenvironment does not exist in other major organs except tumor tissue. Thus, the cross-linking of C-NP and D-NP would not occur in other major organs, which was consistent with the comparable accumulation of S-NP/C-NP and D-NP/C-NP in liver, kidney, spleen, lung and heart (Supplementary Fig. 14).

To demonstrate the stability of C-NP and D-NP in blood, we incubated D-NP and C-NP in PBS solution containing 10% FBS. As shown in Supplementary Fig. 9, the C-NP, D-NP and the D-NP/C-NP formulations all shown good colloidal stability.

Supplementary Fig. 9 D-NP, C-NP and the mixed D-NP/C-NP were incubated with PBS solution containing 10% FBS for different periods, and the size variation were examined by the Zetasizer Nano ZS Instrument. Data were presented as the mean \pm s.d. (n = 3).

Comment 4: Also how did the drug get released? Extracellularly or intracellularly? Releasing kinetics vs. particle clearing kinetics? More characterization should be provided.

Response: The release profiles of encapsulated BB94, DOX, NLG919 and BLZ945 were provided in Supplementary Fig. 15 and Supplementary Fig. 22. And, these encapsulated drugs showed a gradual and sustained release behavior. Meantime, according to the extracellular/intracellular BB94 quantitative examination (Supplementary Fig. 18d-f), it can be concluded that most of injected BB94 were distributed outside tumor cells in the D-NP_{BB94}/C-NP_{BB94} and achieved extracellular release eventually.

Supplementary Fig. 15 The release behavior of encapsulated BB94 from S-NP_{BB94}/C-NP_{BB94} and D-NP_{BB94}/C-NP_{BB94} under different pH environment. Data were presented as the mean \pm s.d. (n = 3).

Supplementary Fig. 22 The release behavior of **a** DOX, **b** NLG919 and **c** BLZ945 from S-NP_{DOX}/C-NP_{BLZ/NLG} and D-NP_{DOX}/C-NP_{BLZ/NLG} under acidity environment (pH 6.5). Data were presented as the mean \pm s.d. (n = 3).

In addition, pharmacokinetics curves of these nanoparticles were also provided in Supplementary Fig. 21. We thank this reviewer for his/her valuable suggestion.

Supplementary Fig. 21 *In vivo* pharmacokinetic profiles after intravenous injection of Cy5-labeled D-NP, S-NP and C-NP. Data were presented as the mean \pm s.d. (n = 3).

Comment 5: In Fig. 7 experiments, indicate when those flow cytometry data were collected. Provide experimental details on this part, like cell numbers and gating information. Line 556 does not provide citation for previous report.

Response: Thanks for your reminding. The flow cytometry data were collected at the end of treatment in Fig. 6c (Line 339). The experimental details including cell numbers and gating information were showed in the Supplementary Fig. 25 in the revised manuscript (Line 341 to line 342). The previous report on line 556 has been cited as Ref. 50 in the revised manuscript (Line 604).

Comment 6: Line 282-284: what is the definition of tumor inhibition rate? Clarify how they are calculated.

Response: The tumor inhibition rate was calculated by the formula of $(V_p - V_t)/V_p \times 100\%$, the V_p represent the final volume of tumor in PBS control group and the V_t represent the final volume of tumor at the end of different treatment group. This calculated method has been described in the revised manuscript (Line 592 to line 595).

Reviewer #3 (Remarks to the Author):

In this manuscript, the author developed a tumor acid triggered bioorthogonal system by cysteine group and 2-cyanobenzothiazole group modified nanoparticles. This technology can be used as drug depots, increasing the retention and enrichment of anticancer drugs at the tumor site. The experiments and arguments in this manuscript were well organized. Thus, I enthusiastically support the acceptance of this manuscript after addressing the following issues:

Comment 1: In Line 158, the author mentioned that "Quantitative region-of-interest (ROI) analysis showed that the fluorescence intensity of the tumor tissue of mice treated with D-NP/C-NP was approximately 2.6-fold higher than that in the S-NP/C-NP group (Fig. 3c)". We wonder that whether "2.6 fold higher" was efficient enough compared with other previous reported bioorthogonal reactions that also used for increasing the retention of anticancer drugs?

Response: In the Fig. 3c, the tumor tissue fluorescence treated with D-NP/C-NP was approximately 2.6-fold higher than that in the S-NP/C-NP group at 96 h post-administration. In addition, according to the suggestion of Reviewer 2, the quantitative drug accumulation was further determined, and the extracellular drug contents in tumor of D-NP/C-NP group were 2.21- and 3.88-folds higher than S-NP/C-NP group at 24 and 48 h post-administration (Supplementary Fig. 18), respectively.

Unlike our study, several groups have used intracellular enzyme-triggered (Ai et al., *Nat. Commun.* 2016, 7, 10432; Ruan et al., *ACS Nano* 2016, 10, 10086-10098) bioorthogonal reactions of nanoparticles to increase the retention of anticancer drugs. Ruan et al. shown that the AuNPs-A&C for 24 h had the highest percentage injected dose per gram glioma (%ID) of 7.52%, which was ~2.3-fold that the control formulation AuNPs-PEG (3.22%, in Figure 5H in *ACS Nano* 2016, 10, 10086-10098).

Comment 2: The experimental description of Figure 4 was not clear enough. Whether BB94-loaded D-NP/C-NP was used for a subcutaneous 4T1 tumor or an orthotopic 4T1 tumor? Why there was no tumor growth curve of the primary tumor? The results in figure 4d-f were analyzed for which kind of tumor, the primary tumor or the metastasis in lungs?

Response: Thanks for your kind reminder. The orthotopic 4T1 breast tumor model was established to evaluate the anticancer activity of BB94-loaded D-NP/C-NP, the description was added in the revised manuscript (Line 220) and Figure 4 caption (Line 238). The results in Figure 4d-f were analyzed for primary tumor, which were marked in the caption of Figure 4 (Line 244 and line 246).

The BB94 is a potent extracellular broad-spectrum MMP inhibitor, and its major function is anti-tumor metastasis. Thus, we did not show the growth curve of the primary tumor previously. According to your valuable suggestion, it was provided in the revised manuscript (Supplementary Fig. 19).

Supplementary Fig. 19 The average growth curves of orthotopic 4T1 breast tumor after different treatment in Fig. 4.

Data were presented as the mean \pm s.d. (n = 5). Statistical significance was calculated via one-way ANOVA with a Tukey post hoc test. * $p < 0.05$.

Comment 3: *The drug models in figure 6 are D-NP (D-NP DOX) and C-NP (C-NP NLG&BLZ). Why was DOX loaded in D-NP alone, while NLG919 and BLZ were co-delivery in C-NP? In addition, whether the use of PLGA-PEG as the backbone for a drug carrier has limitations in the type of drug loading (such as hydrophilic and hydrophobic), and how to adopt strategies to expand the drug loading range by such bioorthogonal reactions in the future, it is recommended to be added in the Discussion.*

Response: We used D-NP to encapsulate DOX and C-NP to co-encapsulate NLG919 and BLZ, our purpose is just to verify the delivery capability of multiple agents. And, all the three drugs were hydrophobic and eventually encapsulated in the PLA core either using D-NP or C-NP. We agree with the reviewer's opinion that the use of PLGA-PEG or PLA-PEG was suitable for hydrophobic drugs. To address this issue, an alginate nanogels system with bioorthogonal in situ assembly capability is being developed by our group, which could be used to encapsulate hydrophilic drugs. The discussion was added in the revised manuscript (Line 442 to line 448). We thank this reviewer for his/her valuable suggestion.

Comment 4: *In figure 7, D-NP (D-NP DOX) and C-NP (C-NP NLG&BLZ) induced an effective anticancer immune response. To further demonstrate the long-term antitumor efficiency, the mice survival experiment of these drugs is suggested to be added.*

Response: According to your suggestion, the mice survival experiment (Fig. 6f) and the description (Line 321 to line 324) was added in the revised manuscript.

Fig. 6f Survival rates of mice treated in different groups, n = 8 biologically independent animals. The statistical significance of the survival was calculated via log-rank (Mantel-Cox) test. **** $p < 0.0001$. G1: PBS, G2: DOX/BLZ945/NLG919, G3: D-NP_{DOX}, G4: D-NP_{DOX}/C-NP, G5: S-NP_{DOX}/C-NP_{BLZ&NLG}, G6: D-NP_{DOX}/C-NP_{BLZ&NLG}.

Comment 5: The author proposed the concept of using bio-orthogonal reactions to develop drug depots. We noticed that D-NP/C-NP could load different kinds of fluorescent dyes (DiI, DiD) and drugs (BB94, DOX, NLG919, BLZ). However, all data supported the argument "enhanced drug retention in tumor site" were according to the results of fluorescent dyes. We suggest the author add another kind of real drug model, which could measure its precise content in tumor by fluorescence intensity or ICP.

Response: We thank this reviewer for his/her valuable suggestion. We chose platinum prodrug as a drug model to achieve precise measurement by ICP-MS (Supplementary Fig. 14). The drug contents in the tumor of D-NP/C-NP group were 1.72- and 2.77-folds higher than S-NP/C-NP group at 48 and 96 h post injection (Supplementary Fig. 14g), respectively. The result was well consistent with the results of fluorescent dyes in Fig. 3c and Fig. 5d.

REVIEWERS' COMMENTS

Reviewer #1 (Remarks to the Author):

The authors have supplemented additional experiments and the raised points have been addressed appropriately. I recommend publication of this manuscript in Nature Communications.

Reviewer #2 (Remarks to the Author):

The revision clarified most of the concerns with new results. However, further attentions are required to the following issues:

1. On the NP bio distribution: the authors carried out an experiment by using the nanoparticles loaded with platinum prodrug and ICP-MS measurement. However, since platinum prodrug is released from the nanoparticles, quantitation of platinum does not necessarily report the amount of nanoparticles at the tissue location. This should be done with covalently labels nanoparticles to accurately quantify the biodistribution.
2. On the cross-link efficiency estimate: From the definition, it seems they independently quantify the amount of S-NP in control tumors. The estimated efficiency was shown to increase over time, but this may be due to the loss of S-NP over the time, rather than more cross linking product formation. Did the authors directly quantify the amount of the cross-linked product in tumor? This would provide much stronger evidence.

Reviewer #3 (Remarks to the Author):

The authors have answered all the questions carefully, all of our concerns have been addressed in the revised manuscript. Thus, we enthusiastically suggest the acceptance of this work.

List of changes and responses to the reviewers' comments

Reviewer #1 (Remarks to the Author):

The authors have supplemented additional experiments and the raised points have been addressed appropriately. I recommend publication of this manuscript in Nature Communications.

Reviewer #2 (Remarks to the Author):

The revision clarified most of the concerns with new results. However, further attentions are required to the following issues:

Comment 1: *On the NP biodistribution: the authors carried out an experiment by using the nanoparticles loaded with platinum prodrug and ICP-MS measurement. However, since platinum prodrug is released from the nanoparticles, quantitation of platinum does not necessarily report the amount of nanoparticles at the tissue location. This should be done with covalently labels nanoparticles to accurately quantify the biodistribution.*

Response: We thank this reviewer for his/her valuable suggestion. To further study the NP biodistribution, we used Cy5 covalently label D-NP/C-NP or S-NP/C-NP to quantify the biodistribution (Supplementary Fig. 15). According to the Supplementary Fig. 15a and 15b, D-NP/C-NP and S-NP/C-NP shown similar accumulation in major organs including heart, liver, spleen, lung and kidney. In tumor tissue, the accumulation of D-NP/C-NP was much higher than that of S-NP/C-NP at 48 h to 96 h post injection, which is well consistent with the ICP-MS measurement results in Supplementary Fig. 14.

Supplementary Fig. 15 The biodistribution of Cy5-labeled D-NP/C-NP and S-NP/C-NP in major organ (a, b) and tumor (c) at 48 h or 96 h i.v. post injection.

Comment 2: *On the cross-link efficiency estimate: From the definition, it seems they independently quantify the amount of S-NP in control tumors. The estimated efficiency was shown to increase over time, but this may be due to the loss of S-NP over the time, rather than more crosslinking product formation. Did the authors directly quantify the amount of the cross-linked product in tumor? This would provide much stronger evidence.*

Response: We understand the reviewers' concern about the direct quantification of the amount of the cross-linked product in tumor. We tried to isolate cross-linked nanoparticles from tumor tissues. However, efficient respective separation of the cross-linked and uncorss-linked nanoparticles from the tumor tissue is very hard, because there are a large amounts of tumor cells, immune cells, and extracellular matrix. Considering the similar properties and pharmacokinetics curve of S-NP/C-NP and D-NP/C-NP, we speculated that their extravasation from blood vessels into tumor tissue was comparable. In addition, another speculation is that cross-linked product is restricted in tumor tissue.

Therefore, an equation about in situ crosslinking efficiency was given in the previous response. However, according to your comment, we realized that the previous equation did not take into account the loss of uncross-linked nanoparticles from the tumor site. Therefore, we used the curve of extracellular NP in Supplementary Fig. 19a and 19b to probably calculate the crosslinking efficiency of D-NP/C-NP in tumor tissue by the following equation:

$$\text{Crosslinking efficiency (CE\%)} = \left(\frac{\text{AUC}_{D\text{-NP/C-NP}} - \text{AUC}_{S\text{-NP/C-NP}}}{\text{AUC}_{D\text{-NP/C-NP}}} \right) * 100\%$$

Where $\text{AUC}_{D\text{-NP/C-NP}}$ and $\text{AUC}_{S\text{-NP/C-NP}}$ represent the area under curve of extracellular contents of D-NP/C-NP and S-NP/C-NP nanoparticles in Figure R1a, respectively. The $\text{AUC}_{S\text{-NP/C-NP}}$ were used to probably calculate the uncross-linked product of the D-NP/C-NP group after injection; and the value of $\text{AUC}_{D\text{-NP/C-NP}}$ minus $\text{AUC}_{S\text{-NP/C-NP}}$ represent the cross-linked product of the D-NP/C-NP group. Thus, the above equation was used to probably calculate the crosslinking efficiency. And, according to the equation, the crosslinking efficiency of D-NP/C-NP in tumor tissues at different times are shown in Figure R1b.

Figure R1. **a** The extracellular NP contents curve of Cy5-labeled D-NP/C-NP and S-NP/C-NP according to Supplementary Fig. 19a and 19b. **b** The NP crosslinking efficiency in tumor tissues calculated by AUC at different times.

Reviewer #3 (Remarks to the Author):

The authors have answered all the questions carefully, all of our concerns have been addressed in the revised manuscript. Thus, we enthusiastically suggest the acceptance of this work.